# Stratospheric wave driving events as an alternative to sudden stratospheric warmings

Thomas Reichler[1], Martin Jucker[2,3]

[1]Department of Atmospheric Sciences, University of Utah, Salt Lake City, 84112, USA
[2]Climate Change Research Center, the University of New South Wales, Sydney, NSW, Australia
[3]Australian Research Council Center of Excellence for Climate Extremes, Sydney, NSW, Australia

*Correspondence to*: Thomas Reichler (thomas.reichler@utah.edu)

**Abstract.** Natural variations in the strength of the northern stratospheric polar vortex, so-called polar vortex events, help to
improve sub-seasonal to seasonal (S2S) predictions of winter climate. Past research on polar vortex events has been largely focused on sudden stratospheric warming events (SSWs), a class of relatively strong weakenings of the polar vortex. Commonly, SSWs are defined when the polar vortex reverses its climatological wintertime westerly wind direction. In this study, however, we use an alternative definition, based on the weighted time-integrated upward wave activity flux at the lower stratosphere. We use a long control simulation with a stratosphere-resolving model and the ERA5 reanalysis to
compare various aspects of the wave activity definition with common SSWs over the Arctic. About half of the wave events are identical to common SSWs. However, there exist several advantages for defining stratospheric weak extremes based on wave events rather than using the common SSW definition: the wave activity flux definition captures with one criterion a variety of different event types, lengthens the prediction horizon of the surface response, and can be more meaningfully applied over the Southern Hemisphere. We therefore conclude that the wave driving represents a useful early indicator for
stratospheric polar vortex events, which exploits the stratospheric potential for creating predictable surface signals better than common SSWs.

## 1  Introduction

The polar vortex is the dominating circulation feature of the northern high latitude wintertime stratosphere. The vortex undergoes pronounced intraseasonal fluctuations in strength (Christiansen, 1999; Kuroda and Kodera, 2001), which we
broadly refer to as polar vortex events. The events are of interest because they persist for several weeks and couple downward into the troposphere to influence surface weather (Baldwin et al., 2003). Knowledge about the events therefore improves tropospheric predictions on subseasonal-to-seasonal (S2S) time scales (Sigmond et al., 2013; Scaife et al., 2021; Domeisen et al., 2020c).

Sudden stratospheric warmings (SSWs) represent the most extreme and best studied example of polar vortex events (Scherhag, 1952; Baldwin et al., 2021). During SSWs, the polar vortex decelerates over the course of a few days and warms at its inner core. This is typically followed by a negative polarity of the annular mode at the surface and a southward shift of the tropospheric mid-latitude jet that lasts for up to two months (Kidston et al., 2015). SSWs also influence the photochemistry of the ozone layer (Mclandress and Shepherd, 2009) and increase the amount of stratospheric ozone (Hong and Reichler, 2021). Minor SSWs are usually distinguished from major SSWs. Major SSWs are the most extreme events, in which the vortex completely breaks down and reverses its climatological wintertime westerly direction. Minor SSWs are less intense, with vortex winds that remain westerly over the course of an event (e.g., Labitzke, 1981). The stratosphere also undergoes prolonged periods with a much stronger than normal vortex, so-called strong vortex or vortex intensification events. These events develop more gradually than weak vortex events, but the meteorological changes associated with them are more or less opposite to that of weak vortex events (Limpasuvan et al., 2005; Baldwin and Dunkerton, 2001).

Different methods have been proposed for the detection of polar vortex events; the papers by Palmeiro et al. (2015) and Butler et al. (2015) give excellent overviews. Most of the methods have in common that some property (e.g., temperature, zonal wind, or geopotential height) of the polar vortex is used, either in terms of an absolute threshold, a pattern, a gradient, or a tendency. Birner and Albers (2017), for example, use the tendency of the zonal mean flow to better capture "the explosive dynamics of these events". The most common definition, however, is based on a reversal of the zonal-mean zonal wind of the polar vortex at mid-stratospheric levels (at 60°N and 10 hPa) (Charlton and Polvani, 2007) (hereafter: CP07). The wind reversal is significant because it represents the complete destruction of the vortex and sets an important condition for wave propagation: easterly winds inhibit the upward propagation of planetary-scale waves (Charney and Drazin, 1961) and is necessary for critical layer interaction (Matsuno, 1971). Arguably, this is important for the intense nature of SSWs and their downward influence on the troposphere. Because of the reversal criterion, the events captured by the CP07 definition are all major SSWs, and for the remainder of this study, we refer to these events simply as SSWs.

The extreme nature of SSWs is probably an important reason for why they have been studied so intensely in the past. However, to date it is unclear how effective the CP07 definition is in capturing events with a downward influence from the stratosphere to the troposphere, which is one of the main reasons for studying polar vortex events in the first place. For example, a study by Sigmond et al. (2013) found that SSWs were followed only in 2/3 of the investigated cases by the expected negative Northern Annular Mode (NAM) (Baldwin and Dunkerton, 2001) at the surface. Another downside of the CP07 definition is that it is based on a fixed threshold and, as long as the zonal wind reverses, the definition also detects the perturbation of a climatologically weak vortex that presumably has a relatively small surface impact. Similarly, events that do not cross the threshold but that nevertheless may have a strong surface impact remain undetected by the fixed threshold definition. In addition, the frequency of SSWs simulated by a model is likely to be affected by biases in the strength of the polar vortex if a fixed threshold criterion is used (e.g., Kim et al., 2017).

The CP07 definition has more shortcomings. For example, climate-change-related long-term trends in the strength of the polar vortex (Karpechko and Manzini, 2017) may change the number of SSWs, even when the stratospheric variability remains unchanged (Mclandress and Shepherd, 2009; Kim et al., 2017). In addition, the CP07 definition is not oriented on the dynamical causes that precede the events but instead on their stratospheric effect. This may be relevant for the prediction of polar vortex events in the context of S2S applications, since a more cause-centered approach could lengthen the relatively short 1-2 week-long predictability limit for polar vortex events (Domeisen et al., 2020b).

The purpose of this study is to present and evaluate an alternative definition for polar vortex events, which avoids some of the shortcomings of the CP07 definition. The new definition is based on the upward planetary-scale wave activity flux at 100 hPa, or equivalently, the poleward eddy heat flux, which is often referred to as the "stratospheric wave driving" (Newman and Nash, 2000)[1]. It is well known that the wave driving plays an essential role for the stratospheric circulation. This recognition goes back to the Matsuno model for SSWs (Matsuno, 1971), providing the first dynamical explanation for SSWs in terms of the interaction of vertically propagating planetary-scale waves with the zonal flow. Later, Newman et al. (2001) used the transformed Eulerian mean framework (Andrews et al., 1987) to further clarify from reanalysis the essential role of the wave driving for the winter stratospheric circulation and temperatures. The seminal work by Matsuno was also followed by numerous modelling studies, which investigated the generation of SSWs by planetary-scale waves (e.g., Holton and Mass, 1976; Reichler et al., 2005). A statistical analysis by Jucker and Reichler (2018) showed that the wave driving increases the probability of SSWs within the following three weeks and thus helps predicting SSWs. Other studies linked periods of reduced wave driving and the resulting absence of wave-mean flow interaction to the formation of strong vortex events (e.g., Limpasuvan et al., 2005; Lawrence et al., 2020; Polvani and Waugh, 2004).

Using the wave driving for the detection of polar vortex events is not new. Polvani and Waugh (2004) used a threshold criterion based on the 40-day averaged upward flux at 100 hPa to define events. The NAM composites that followed the events (their Fig. 4) looked very similar to the famous "dripping paint" plots by Baldwin and Dunkerton (2001), demonstrating that the wave activity flux is an indicator for subsequent polar vortex events. However, there remain many open questions. For example, a systematic comparison between SSWs and wave driving events has not been performed, leaving it unclear how robustly the polar vortex and the surface respond to the wave driving in comparison to SSWs. Also, a statistical characterization of wave driving events and how they compare to SSWs is still missing.

Another question concerns the exact wave driving criterion that should be used to define the events. Most previous studies agreed that the wave activity flux in the lower stratosphere (100 hPa) is important, since at this level the filtering of the

---

[1] The term wave driving is perhaps somewhat misleading because it is the convergence of the wave flux and not the flux itself that drives the polar vortex. However, in the literature, wave driving is often used to refer to the flux, and we keep with this tradition.

waves at the tropopause is no longer an issue (Chen and Robinson, 1992). We note that the flux at 100 hPa should not be simply interpreted as a wave propagation from the troposphere into the stratosphere. Cámara et al. (2017) showed that only 1/3 of the wave flux variance at 100 hPa can be explained from the flux in the upper troposphere (300 hPa). They argued that the 100 hPa level is well above the extratropical tropopause and thus already under the considerable influence of stratospheric processes. In the context of our study, however, the exact source for the wave activity flux is less important. Several previous studies also indicated that daily values of the wave activity flux are less important than the time-integrated values. For example, Newman et al. (2001) showed that the 45-day accumulated wave driving at 100 hPa in middle to late winter was highly correlated with the subsequent March polar stratospheric temperatures. Further, observational studies showed that individual wave driving events tended to last for one to two weeks (Randel et al., 2002), and that perturbations of the polar vortex on a given day were not so much related to the instantaneous upward wave activity as to its integral over several weeks prior to that date (Polvani and Waugh, 2004). Similarly, Sjoberg and Birner (2012) found that wave driving with a relatively long time-scale (> 9 days) was more effective in generating SSWs than a strong but short pulse of wave activity. Therefore, and as we will explain in more detail below, we consider in our study a weighted time-integral of the wave driving at the lower stratosphere (100 hPa) to define polar vortex events. As mentioned above, the 100 hPa level is probably the most common way to measure the wave activity flux that enters the stratosphere. The 100 hPa level is well above the extratropical tropopause (ca. 200 hPa), but also low enough to create some extra lead time between wave activity flux and vortex perturbation. This time of ca. 4 days is needed for the waves to propagate from the lower to the middle stratosphere and undergo wave–mean flow interaction (Horan and Reichler, 2017).

We examine a modern reanalysis data set and a long simulation with a realistic coupled stratosphere-resolving model to expand and clarify the findings from previous studies. This is achieved by consequently comparing the results from the wave driving definition against the Charlton and Polvani SSW definition. Because the atmosphere is chaotic, we describe most of our results in a statistical sense, which is greatly facilitated by using the data from the long model simulation. The large number of events captured by this simulation allows detailed examination of distributions and sub-samples of specific events. Most of our analysis is complemented by an investigation of the reanalysis to provide a baseline for the observed atmosphere. Further, we mostly focus on events preceded by anomalously positive wave driving because of their similarity to SSWs, but where practical, we also include events with a reduced wave driving.

Section 2 of this paper starts with explaining our data and the model simulation. Section 2 goes on to describe how we define the wave driving events and we explain our statistical methods. The results in Section 3 have seven parts. First, we validate the model in terms of quantities that are relevant to this study. We then examine the sensitivity of the new definition to the minimum wave driving threshold. Next, we describe the typical life cycle of wave driving events and examine the evolution of the sea level pressure anomalies prior to the events. We continue by describing the occurrence of past wave driving events in the observations. We then investigate the probabilistic relationship between the wave driving, the polar vortex winds, and the sea level pressure, and we present the seasonality of wave driving events in terms of their frequency and surface

response. The result section concludes with presenting the spatial sea level pressure response from the various event types. Together, our results illustrate that defining polar vortex events solely from information about the preceding wave driving works surprisingly well and has a number of advantages over the CP07 SSW approach. The paper concludes with a discussion in Section 4.

## 2 Data and Methodology

Daily observational estimates are derived from the ECMWF reanalysis v5 (ERA5) (Hersbach et al., 2020) over the period 1979-2020. The reanalysis was downloaded at hourly intervals and a 0.25° resolution, and then averaged to daily values and interpolated to a 1° grid. Daily simulation data are derived from a nearly 10,000 year-long present-day control run with HI-CM2.1 (henceforth: "the model"), the stratospheric resolving version of the coupled climate model CM2.1 from GFDL (Delworth et al., 2005). The model has 48 vertical levels (Staten and Reichler, 2014), twice as much as the original CM2.1, a model lid at 0.002 hPa (ca. 92 km), and a 144 x 90 global horizontal grid (ca. 2° x 2°). Greenhouse gases, ozone concentrations, and other external forcings of the simulation were prescribed at 1990 levels and held constant through time. The first 1000 years of the simulation are discarded to reduce initial spin-up problems.

We distinguish two types of polar vortex events. The first is SSWs, detected according to the CP07 definition (Charlton and Polvani, 2007). An SSW occurs when the daily zonal mean zonal wind at 10 hPa and 60°N ($u_{1060}$) shifts from westerly to easterly (i.e., the onset or central date $t_0$) between 1 November and 31 March, provided that afterwards the vortex returns to westerly for at least 10 consecutive days before 30 April. Multiple SSWs per season must be separated by at least 20 consecutive days of westerlies. For each SSW, we also determine $u_{min}$, the associated minimum $u_{1060}$ during the 10-day period after onset, $du_{min}$, the anomaly of $u_{min}$ with respect to the daily $u_{1060}$ climatology, and $\sum F_Z'$, defined in the following section.

The second event type is so-called wave-driving events. Their definition is based on the lower stratospheric (100 hPa) wave activity flux, given by daily values of the vertical component $F_P$ of the quasi-geostrophic Eliassen-Palm (EP) (Eliassen and Palm, 1961) flux. We use the vertical EP-flux component in pressure coordinates (Andrews et al., 1987), which is given, using standard notation, by $F_P = a \cos \phi \, f \frac{\overline{v'\theta'}}{\overline{\theta}_p}$. We calculate $F_P$ for all waves, average using latitude weighting from 20°N to 90°N, and scale by -1 x $10^5$ kg·m·s$^{-4}$ to arrive at $F_Z$. The scaling, denoted as one wave driving unit (1 WDU), nondimensionalizes the flux, creates magnitudes that are close to unity, and ensures that a positive sign means upward propagation. We then normalize $F_Z$ by removing the daily climatology of $F_Z$ (taken from all available years) and dividing by the daily standard deviation (3.17 WDU for ERA5, 3.49 WDU for the model) (Fig. 1) to arrive at $F_z'$.

The search for "Positive Wave Driving events" (PWDs) starts each winter on 1 November by setting both the current time index $t_0$ and the time index of a prior event $t_{prior}$ to this day. We then advance $t_0$ at daily intervals and calculate each time the

sum of the weighted $F_z'$ (daily $F_z'$ values can be of either sign) between $t_0$ and $t_{prior}$. The weights are a decaying exponential with an e-folding time of $\tau = 50$ day, so that values closer to $t_0$ carry more weight than values further away from it. This can be written as

$$\Sigma F_Z' = \sum_{t=t_{prior}}^{t_0} \exp\left(\frac{t - t_0}{\tau}\right) F_Z'(t) .$$

Note that $\Sigma F_Z'$ has units of (stddev · day), or simply (day). The rationale behind the uneven weighting is that the memory of past wave driving diminishes with time as the vortex tends to relax to climatology. By experimentation we found that it is important to use a long enough e-folding time $\tau$: a shorter $\tau$ selects events that tend to be preceded by stronger negative stratospheric NAM anomalies and followed by weaker surface responses. The $\tau = 50$ day of our study is also similar to previous studies (Polvani and Waugh, 2004; Newman et al., 2001).

Using the above definition, a PWD is detected when the accumulated wave driving reaches a certain critical threshold, i.e., when

$$\Sigma F_Z' \geq \Sigma F_{Z\,crit}' .$$

As we explain in more detail in section 3.2, in most of our study we use an empirically determined threshold of 12.9 day for the model data and of 12.3 day for the ERA5. We further advance $t_0$ until the daily $F_z'(t_0)$ becomes negative. The $t_0$ when this happens determines the end of the wave event and the final onset date of the PWD. We then search for additional PWDs by setting $t_{prior} = t_0$ and repeating the above-described summation. As for SSWs, multiple PWDs in the same winter must be separated by 20 or more days, but there is no extra requirement for the sign of $F_z'$ or $u_{1060}$ during this period. "Negative Wave Driving events" (NWDs) are defined just like PWDs, except that $\Sigma F_{Z\,crit}'$ is negative and the sign of the inequalities is reversed.

We end the search for wave driving events on 1 June of the next year, which means that PWD onset dates in April or even later are a possibility. In practice, however, PWDs in April are rare, and in the model only two events (0.038 % of all events) were detected at the beginning of May.

Similarly to SSWs, we determine and save for each event $u_{min}$, the minimum $u_{1060}$ in the ±10 day neighborhood of the onset, $du_{min}$, the anomaly of $u_{min}$ with respect to the daily $u_{1060}$ climatology, and the accumulated wave driving $\Sigma F_Z'$ prior to the event. Table 1 lists some of the $\Sigma F_Z'$ statistics for SSWs, PWDs, and NWDs. Unsurprisingly, in the mean, SSWs and also PWDs have positive $\Sigma F_Z'$ values, but the average $\Sigma F_Z'$ for PWDs is larger than that for SSWs. Some SSWs are even preceded by a negative $\Sigma F_Z'$.

Since certain SSWs and PWDs mark the same event, we define a common event (COM) as a PWD which has an onset date that is within ±20 days of an SSWs. Likewise, exclusive PWDs (EX-PWDs) and exclusive SSWs (EX-SSWs) are events that are either only a PWD or only an SSW.

We use daily surface temperature, precipitation, and sea level pressure (SLP) to describe the surface conditions that follow stratospheric events. Daily time series of the anomalous SLP averaged over the northern Polar Cap ($\overline{slp}^{PC}$) and the North Pacific ($\overline{slp}^{NP}$) are derived from averaging SLP (using latitude weighting) over (60° - 90°N and all longitudes) and (35° - 60°N, 160°E - 120°W), respectively. We further calculate $\overline{slp}^{PC,0-59}$, the time averaged $\overline{slp}^{PC}$ from days 0 - 59 after the onset of an event to provide an approximate measure of the integrated strength of the surface impact of an event.

We analyze the downward coupling of vortex events in terms of the standardized northern polar-cap-averaged (60° - 90°N) geopotential height anomalies. The outcome is similar to the empirical-orthogonal-function-based approach of the Northern Annular Mode index (Baldwin and Thompson, 2009). A two-sided Student $t$-test at $p < 0.05$ is used to calculate the statistical significance of composite anomalies and the length of confidence intervals.

A daily Niño 3.4 index is constructed from the anomalous surface air temperature averaged over the Niño 3.4 region (5°S - 5°N, 120° - 170°W). An annual (winter) Niño 3.4 index is then derived from taking a 6-month average of the daily index centered on February 1. The phases of ENSO are identified as follows: when the annual Niño 3.4 index of a particular winter exceeds either the upper or the lower quartile of the distribution of the annual index, an El Niño or La Niña event is defined. The thresholds for the two quartiles are ±0.50 K for the ERA5 and ±0.89 K for the model, reflecting the model's higher ENSO variability ($\sigma = 1.3$ K) (Wittenberg et al., 2006) compared to the ERA5 ($\sigma = 0.7$ K).

## 3 Results

### 3.1 Model validation

We begin by validating the model against ERA5. Multiple previous studies have already demonstrated that HI-CM2.1 and its low-top companion (CM2.1) produce realistic simulations of the troposphere (Reichler and Kim, 2008), stratosphere (Horan and Reichler, 2017; Staten and Reichler, 2014; Jucker et al., 2021), and ocean (Gnanadesikan et al., 2006; Wittenberg et al., 2006). Here, we investigate two quantities that are important for this study: the zonal-mean zonal wind at 60°N and 10 hPa ($u_{1060}$) and the upward wave activity flux at 100 hPa ($F_Z$). The model simulates their climatological seasonal cycle and their interannual standard deviation well (Fig. 1a and 1b). The model also captures the reanalysis well in terms of the daily distribution of $\overline{F_Z}$ during northern winter (DJFM) (Fig. 1c). As described in the upcoming sections, additional confidence into the model's performance is derived from the good agreement between the simulated (58%) and ERA5-derived (62%) SSW frequency and the similarity of the SLP response to polar vortex events in the two datasets. Note that in this paper the event frequency is given in events per year multiplied by 100 (%).

## 3.2 Sensitivity of wave driving events to $\overline{F_{Z}'}_{CRIT}$

We next investigate how sensitive PWDs are to the choice of $\sum F_{Z\,crit}'$, both in terms of the event frequency and the SLP response ($\overline{slp}^{PC,0-59}$, see Section 2). The red curves in the top panels of Fig. 2 demonstrate that with increasing $\sum F_{Z\,crit}'$, the model simulated PWD frequency (left) decreases and the SLP response to PWDs (right) increases. The horizontal black lines are the corresponding outcomes for SSWs, i.e., 58% and 1.7 hPa, respectively. In terms of the frequency (Fig. 2a), the curves for PWDs and SSWs intersect at $\sum F_{Z\,crit}' \sim 12.9$ days. Therefore, we focus on this threshold for the rest of this study when defining PWDs or NWDs. At this threshold, PWDs create a somewhat stronger SLP response (~2.0 hPa) than SSWs (~1.7 hPa), and, as can be seen by the shading in Fig. 2b, this difference is statistically significant. We also note that strong EX-PWDs, i.e., polar vortex events with $\sum F_{Z}' > 16$ WDU but which are missed by the SSW definition, create on average a stronger SLP response than SSWs. Conversely, EX-SSWs, i.e., polar vortex events that are not classified as PWDs, create a rather weak SLP response. It is further of interest that at $\sum F_{Z\,crit}' \sim 12.9$ day, about half of all PWDs are also SSWs (Fig. 2a). The bottom panels of Fig. 2 demonstrate that, overall, ERA5 lead to fairly similar results as the model. But due to the smaller sample size, the results are much noisier and the SLP difference between PDWs and SSWs is not significant.

## 3.3 Life cycles of stratospheric events

Past studies used the concept of composite-mean life cycles to illustrate the typical temporal evolution of stratospheric events (e.g., Baldwin and Dunkerton, 2001). Fig. 3 follows this concept and presents life cycles of SSWs, PWDs and NWDs for (left) the ERA5 and (right) the model. Shown are composites for different quantities, each centered on the onset date $t_0$ (day 0) of the events. The following discussion is focused on the results for the model, but we note that the results for the reanalysis are quite similar.

Figs. 3a-b show the composite evolution of $F_z'$ at 100 hPa. Both PWDs (red) and SSWs (black) are preceded by pronounced positive wave driving which peaks at about 5-6 WDU, but the wave driving for PWDs is somewhat broader and stronger than for SSWs and starts ~10 days earlier. According to Table 1, PWDs are on average preceded by 39% more accumulated wave driving than SSWs (61 vs. 44 WDU·day). Both PWDs and SSWs are also followed by persistent negative wave driving anomalies, a well-known result (e.g., Hong and Reichler, 2021; Hitchcock and Haynes, 2016). The wave driving for NWDs (blue) is mostly symmetrical but opposite to that of PWDs, but the negative anomalies start somewhat earlier and reach only about -4 WDU at onset.

Figs. 3c-d illustrate the response of the polar vortex in terms of $u_{1060}$ along with the annual climatological cycle of $u_{1060}$ centered on the respective onset dates. While SSWs cross the zero-wind line at onset by definition, $u_{1060}$ for PWDs does not quite reach zero, suggesting that not all PWDs cross the zero-wind line. From the differences between the continuous lines and the dashed lines for the climatology in Figs. 3c-d one can see that PWDs and SSWs are also associated with somewhat negative $u_{1060}$ anomalies as early as 50-60 days before onset, hinting at vortex preconditioning prior to the events (Lawrence

and Manney, 2020). Consistent with the reduced wave driving after onset, the vortex of both SSWs and PWDs becomes anomalously strong starting ~40 days after onset and remains so for the rest of the winter. As can be seen from the time of the $u_{1060}$ zero crossing 80-90 days after the onset of SSWs and PWDs, the stronger-than-normal vortex is then associated with final warmings[2] (Black and Mcdaniel, 2007) that are about 15 days later than climatology, both in the observations (Fig. 3c) and in the model (Fig. 3d) (see also Hu et al., 2014). For NWDs, the vortex first becomes anomalously weak (~5 m/s) starting at ~60 days before onset, then becomes anomalously strong (~15 m/s) due to the reduced wave driving, and it maximizes a few days after onset. Then, the wind anomalies gradually weaken to reach climatology at ~30 days after onset. Afterwards, there is "under-recovery" of the vortex and the FW date of NWDs is also about normal.

Figs. 3e-f show that both SSWs and PWDs are followed by positive $\overline{slp}^{PC}$, corresponding to the negative phase of the North Atlantic Oscillation (NAO) index. The anomalies maximize ~5-10 days after onset and decay almost linearly over a period of 60+ days. It is of note that PWDs are associated with somewhat stronger and more persistent anomalies than SSWs, and that, as mentioned before (Fig. 2b), this difference is statistically significant. We further note that both event types are preceded by strong negative $\overline{slp}^{PC}$ anomalies that start as early as ~20 days before the events. NWDs are preceded by significant negative $\overline{slp}^{PC}$ anomalies starting ca. 50 days before onset, which then briefly turn positive a few days before onset. The $\overline{slp}^{PC}$ anomalies that follow are mostly opposite to that of PWDs, with long-lasting (> 90 days) and strongly negative values.

To understand the spatial structure behind the SLP anomalies that precede the events, we show in Fig. 4 Northern Hemisphere (NH) maps of model-simulated anomalous composite mean SLP at 10-day intervals prior to event onset. The composites only include stratospheric events during neutral ENSO years to avoid the teleconnection influences from ENSO. At lags of -25 and -15 days, the PWD composites show a high-latitude dipole with low pressure over the western hemisphere and high pressure over the eastern hemisphere. The dipole is broadly similar to the tropospheric precursors of vortex weakening events described by Garfinkel et al. (2010) and represents a strengthening of the climatological wavenumber-1 component of geopotential heights. Averaged over the polar cap, these anomalies are predominantly negative, explaining the preceding negative $\overline{slp}^{PC}$ seen in Fig. 3f. Five days before onset, the PWD dipole intensifies and contracts more poleward, with the high over the Euro-Atlantic sector being indicative for blocking (Martius et al., 2009; Barriopedro and Calvo, 2014). At the same time, a north-south dipole emerges over the western North Pacific, somewhat reminiscent of previous observational findings (Cohen and Jones, 2011; Lehtonen and Karpechko, 2016; Dai and Hitchcock, 2021). Five days after onset, the canonical negative NAO pattern develops, largely opposite to the SLP pattern at day -5. This demonstrates that

---

[2] Final warmings (FWs) represent the last weakening of the polar vortex at the end of winter when the seasonal increase of radiative heating over the pole prevents the reformation of the vortex.

tropospheric high-latitude weather undergoes radical changes from the precursor and downward influence of stratospheric events. The SLP patterns for SSWs are like PWDs, but generally weaker in magnitude.

The situation for NWDs is quite different (Fig. 4, bottom). Starting as early as day -25, NWDs are preceded by pronounced SLP anomalies resembling a positive NAO over the North Atlantic sector. There is also a strongly positive anomaly over the North Pacific, which together with the negative one over Iceland reduces the climatological wavenumber-1 pattern and explains the negative wave driving anomaly seen in Fig. 3b. The composite Nino 3.4 index during NWDs is -0.24 K, suggesting that NWDs are somewhat favored by La Nina-like conditions. Note that only events from neutral ENSO years are

included in these composites, but that the Nino 3.4 limits for neutral ENSO are ±0.89 K. The weak La Nina conditions help explaining the persistent positive SLP anomalies over the North Pacific that start long before onset.

The ERA5 reanalysis exhibit somewhat similar but much noisier patterns (not shown) due to the small number of events. We also caution that Fig. 4 only offers limited insight, since the figure merges possibly different precursor types into one overall mean. More work is needed to resolve this issue and better understand the precursors that lead to stratospheric events.

Going back to Fig. 3, the six bottom panels show time-height cross sections of the NAM index. It is quite remarkable that PWDs, which are entirely based on information about the wave driving prior to the events, show a very similar timing and magnitude of the NAM anomalies as SSWs, despite the additional uncertainty of PWDs of how exactly the stratosphere responds to the wave driving. The main difference is that PWDs show a somewhat stronger and more persistent negative surface NAM after onset and a more positive stratospheric NAM ~30 days prior to onset. Note that negative (positive) NAM

anomalies correspond to positive (negative) polar-cap-averaged geopotential height anomalies and are therefore shown in Fig. 3 by reddish (blueish) shading. The more positive preceding NAM hints that PWDs are perhaps more closely related to polar night jet oscillations or vacillations (Kuroda and Kodera, 2001; Christiansen, 1999), quasiperiodic oscillations from the delayed mutual influences between wave activity flux and vortex strength (Birner and Albers, 2017), than SSWs. Another interesting observation is that both SSWs and PWDs are preceded a few days before onset by positive NAM anomalies at the

surface, which are slightly stronger for SSWs than for PWDs. This is consistent with the different $\overline{slp}^{PC}$ evolutions shown in Fig. 3f and Fig. 4. We also note that the PWD-composites of the NAM from the ERA5 (Fig. 4i) are similar to the high heat flux composites by Polvani and Waugh (2004) for the NCEP/NCAR reanalysis (their Fig. 4a) and the "dripping paint" composites by Baldwin and Dunkerton (2001) (their Fig. 2). NWDs (Fig. 3l) before onset are characterized by persistent positive NAM anomalies owing to the reduced wave driving long before onset. NWDs are followed by NAM anomalies that

are quite similar (but opposite) to that of PWDs, but the magnitude is weaker, and the long persistence (>90 days) of the anomalies is quite remarkable.

## 3.4 History of past events

When did wave driving events occur in the real atmosphere, and how do the events compare to SSWs? Fig. 5 answers this by showing the evolution of $u_{1060}$ along with the timing of SSWs, PWDs, and NWDs (triangles) and their accumulated wave activity fluxes (numbers) in the ERA5. Over the 42-year-long period, we detect 26 SSWs, 26 PWDs, and 26 NWDs. Some events, like January 2019 or February 2018, are both SSWs and PWDs, or so-called common events. Using 20 days as the maximum separation distance between SSWs and PWDs, there were 15 common events. In other words, somewhat more than half of the 26 SSWs were preceded by sufficiently strong lower stratospheric wave activity flux to also classify as PWDs. In the model, we also find that roughly half of all SSWs are also PWDs (Fig. 2a, top panel). These numbers are largely consistent with a study by White et al. (2019), which found that 60% of the SSWs in their model are preceded by an extreme wave activity at 100 hPa.

On the other hand, eleven SSWs and eleven PWDs in the ERA5 occurred independent from each other, indicating that there exist important differences between some of the two types of events. SSWs, for example, are not always preceded by strong lower stratospheric upward propagating wave activity flux. Previous work has shown that other factors, like stratospheric internal dynamics (Scott and Polvani, 2004, 2006; Cámara et al., 2019), can also create SSWs. In addition, the stratospheric background state also plays a role, for example by altering the propagation of the waves. This was highlighted by Cámara et al. (2017), who found that a strong wave flux at 100 hPa is not sufficient to produce an SSW and that the ''right'' stratospheric state is also essential. Similar arguments may hold for PWDs.

One important difference between SSWs and PWDs may come from events that occur either very early or very late in winter, when the polar vortex is climatologically weaker. In this case, small amounts of wave driving may be sufficient to create SSWs, but this would not produce a PWD. One such example is the late SSW from 1988 (Fig. 5), which occurred on March 13 (day 72), and which was associated with a slightly negative wave driving ($\sum F_Z' = -1$ day). Another SSW with a notably weak wave driving was 2008 ($\sum F_Z' = 4$ days). Fig. 5 also shows exclusive mid-winter PWDs that were not SSWs, for example during the "decade without SSWs" of the 1990s, or the event from 2016, which was the second strongest PWD ($\sum F_Z' = 25$ days) during the ERA5 period. Overall, the strongest PWD was in February 2009 ($\sum F_Z' = 27$ days), which was also an SSW. Albers and Birner (2014) argued that this event may have been triggered by nonlinear resonant wave amplification (Matthewman and Esler, 2011; Esler and Matthewman, 2011) in the stratosphere, which does not require intense tropospheric wave activity.

In the ERA5, we also find 26 NWDs, events in which sustained amounts of anomalously negative wave driving create a cold and strong polar vortex. Fig. 5 shows that many NWDs occur in close proximity to warm vortex events (e.g., 1981, 1982, 1988, 1995, ...), which may be related to the oscillatory nature of the stratospheric circulation. However, there also exist isolated NWDs, for example the strong vortex of 2020 described by Lawrence et al. (2020). The overall strongest NWD was in 2011 ($\sum F_Z' = -21$ days), followed by 1989 ($\sum F_Z' = -20$ days).

### 3.5 Relationships between wave driving, polar vortex perturbation, and surface response

Although polar vortex events contribute to prediction skill of subseasonal NH winter climate variations, one difficulty is that not every event affects the troposphere (e.g., Karpechko et al., 2017; Jucker, 2016). The reason is that the state of the troposphere during the events plays a role for the characteristics of the surface response (Domeisen et al., 2020a; Oehrlein et al., 2021). When vortex events are not defined from the perturbation of the vortex (i.e., SSWs) but from the wave driving (i.e., PWDs), an additional complication arises. This is related to the chaotic nature of the atmospheric flow and not exactly

knowing how the polar vortex will respond to the preceding wave activity flux, or more precisely, how much of the lower stratospheric wave activity flux converges into the region of the polar vortex. In other words, using the wave driving to define stratospheric events may further increase the already uncertain surface response to polar vortex events.

We use the model data to explore this possibility and present in Fig. 6 the distributions of responses to SSW and PWD events in terms of (a) the perturbation of the polar vortex wind, $du_{min}$, and (b) the polar-cap-averaged SLP anomaly over the

340 0-59 day period following the events. Fig. 6a illustrates that during SSWs (black), the polar vortex decelerates on average by ~31 m/s, with a range of outcomes from ~4 m/s to ~70 m/s. Overall, the situation for PWDs (red) is quite similar, indicating that the above-mentioned uncertainty from not knowing how the vortex will respond to the wave driving is small. Closer inspection shows that the mean vortex deceleration during PWDs amounts to ~29 m/s, somewhat smaller than for SSWs. In addition, there exist some PWDs with a positive vortex perturbation $du_{min}$, but the number of these events is very small.

For practical purposes, the response at the surface is more important than the perturbation of the vortex. Histograms of the surface response ($\overline{slp}^{PC,0-59}$) to the same SSWs and PWDs as in the left panel are shown in Fig. 6b. There is a wide range of responses, from minus 10 hPa to plus 12 hPa, clearly demonstrating how uncertain the surface response to stratospheric events can be. Overall, the two distributions are again very similar. As expected, both are shifted towards positive SLP anomalies, corresponding to the negative phase of the NAO. Compared to SSWs, PWDs create on average a somewhat

stronger mean response (2.0 hPa vs. 1.7 hPa), reduced response spread (3.5 vs. 3.6 hPa), and reduced chance of a negative $\overline{slp}^{PC,0-59}$ (29% vs. 32%).

These results suggest that the response of the polar vortex to the wave driving is not much more variable than the response of the vortex to SSWs (Fig. 6a), and this does not affect much the surface response (Fig. 6b). The main uncertainty of the surface response stems from the downward migration of the stratospheric signal in the presence of strong tropospheric

weather noise, and PWDs and SSWs behave in this respect very similarly. As already seen before (Figs. 2b and 3f), there is indication that PWDs create a somewhat more robust surface response than SSWs, which is consistent with other previous studies that preceding strong upward propagating wave activity is an early indicator for downward propagating SSWs (Karpechko et al., 2017; White et al., 2019).

### 3.6 Seasonality of event frequency and surface response

Next, we explore in Fig. 7 several aspects of event seasonality. As before, this analysis does not include data from the reanalysis as there are too few observed events. Fig. 7a shows how the different event types are distributed over the various months. We first note that, compared to PWDs (red) and NWDs (blue), SSWs (black) exhibit a much narrower distribution which peaks in February. The February peak is not entirely consistent with the observed SSWs which maximize in January (Butchart et al., 2011), but we caution that the number of observed events is too small for such a conclusion (Horan and Reichler, 2017).

Fig. 7a shows additional event types. Most notably, exclusive SSWs (EX-SSWs), SSWs associated with a rather small wave driving ($< \overline{F_{Z}}'_{CRIT}$), are most common in March. This is a time when the vortex is weak and even small amounts of wave driving are able to reverse the vortex. Exclusive PWDs (EX-PWDs) are broken down into events in which the polar vortex does (U-) and does not (U+) cross the zero-wind threshold. U- events are most common in March and April, a timing that suggests that many of the events are final warmings. Some of the April U- events are associated with a complete vortex recovery (not shown), but these events are not SSWs since they are not permitted by the CP07 definition. U+ events, on the other hand, maximize in December and January, when the vortex is strong and requires considerable forcing to break it down. Since the vortex does not reverse in this case, U+ events are comparable to classical mid-winter minor warmings.

The following panels of Fig. 7 are concerned with the strength and seasonality of the surface response. As before, the response is measured in terms of $\overline{slp}^{PC,0-59}$. Fig. 7b goes back to a question raised before, i.e., how many vortex events are "downward propagating". In Fig. 7b this is answered in terms of the percentage of events followed by the expected sign of the polar-cap-averaged SLP anomaly, i.e., positive for SSWs and PWDs and negative for NWDs. The numbers next to the event labels show the outcomes averaged over all months. 71% of all PWDs are followed by the expected positive $\overline{slp}^{PC,0-59}$, with similar numbers for SSWs and NWDs. This outcome is close to what White et al. (2019) found in their model, that ~67% of SSWs that were preceded by extreme lower-tropospheric wave activity were downward propagating in the sense of Karpechko et al. (2017). Of note is the strong decline of the expected response to SSWs towards late winter, which closes in at the critical 50% mark. These late SSWs are frequent (Fig. 7a) but associated with weak surface responses. The likely reason is that dynamically these events are not very active; the climatological vortex is weak during this time of the year and small amounts of wave activity suffice to trigger the SSW criterion. On the other hand, most of the PWDs during this late time of the year show the expected surface response, since by definition they are always associated with a large wave flux activity.

Figs. 7c-d present the average surface response ($\overline{slp}^{PC,0-59}$) by the time of the year. As in Fig. 7b, the SSW response varies strongly by month: it maximizes at 2-3 hPa in mid-winter and declines towards the end of winter. In contrast, the response to PWDs is more moderate (~1-2 hPa) during most months.

EX-SSWs are of particular interest because they are missed by the PWD definition. EX-SSWs during early- and mid-winter create sizeable responses (Fig. 7d) but are not very frequent (Fig. 7a). They are more common during February and March (Fig. 7a) but then their surface response is weak (Fig. 7d). EX-PWDs, on the other hand, have two distinct frequency peaks (Fig. 7a), one from (1) U+ events in mid-winter and another one from (2) U- events in late-winter, both of which create sizeable surface responses (Fig. 7d). Overall, this suggests that PWDs that are missed by the SSW definition (EX-PWDs) are

more relevant for the surface than SSWs that are missed by the PWD definition (EX-SSWs). Fig. 7e further illustrates this by showing the frequency weighted SLP responses to EX-PWDs and EX-SSWs. The area under each curve is a measure for the overall relevance of the events. Only in February are EX-SSWs more relevant than EX-PWDs, mostly because EX-SSWs during this time are so frequent.

## 3.7  SLP response patterns

We conclude our study by examining the surface response that follows the various stratospheric events in terms of the spatial SLP pattern (Fig. 8). Note that Fig. 8 as most of our figures (except Fig. 4) show results for all years, and not just from ENSO-neutral years. As expected, PWDs and SSWs are all associated with a negative phase of the NAO, i.e., higher pressure over the polar cap and lower pressure over the North Atlantic-European sector. In addition, PWDs are associated with negative anomalies over the North Pacific. This is likely related to the teleconnection influence from the El Niño

Southern Oscillation (ENSO) on the climatological Aleutian Low (Horel and Wallace, 1981). The deepening of the low intensifies the planetary wave #1 activity, provides some of the wave forcing needed for PWDs, and overall increases the likelihood for polar vortex events  (Garfinkel and Hartmann, 2008).

The SLP response to NWDs is roughly inverse to that of PWDs, except that the North Pacific SLP anomalies of NWDs are stronger (i.e., +3 hPa vs. -2 hPa). Perhaps, remote forcing from ENSO plays a more important role for NWDs than for

PWDs. This interpretation is supported by the composite Niño 3.4 index of -0.80 K during NWDs and +0.63 K during PWDs. In contrast, the composite Niño 3.4 index during SSWs is only +0.05 K, indicating that in our model ENSO plays almost no role for SSWs. In a separate upcoming paper, we plan to better understand the role of ENSO in influencing polar vortex events and their surface response.

Here, we are mostly interested in the differences between SSWs and PWDs and therefore focus on events that are mutually

exclusive from each other, i.e., EX-SSWs and EX-PWDs. Fig. 8 shows EX-PWDs separately for U+ and U- events. In the model, about half of all SSWs and also half of all PWDs are exclusive events. As mentioned before, EX-SSWs are followed by a quite modest SLP response, weaker than that to U+ or U- events. U+ events are associated with particularly negative SLP anomalies over the North Pacific, presumably because of a strong ENSO influence on these events. U- events are not very frequent, but they create robust positive SLP anomalies over the Polar Cap. U- events occur on average on March 20,

much earlier than the model's mean FW date of April 10 (Fig. 1a). U- events can therefore be seen as early but impactful "dynamical" FWs, which occur during a time when the climatological vortex still requires a substantial wave forcing to

break it down. These FWs should be distinguished from "radiative" FWs, which are simply due to the seasonal increase of the radiative heating over the pole. Furthermore, U- events are preceded by an anomalously strong vortex 1-2 months before onset (not shown), which is consistent with Hu et al. (2014) who showed that winters with a strong polar vortex tend to be followed by early FWs.

Fig. 8 also presents the SLP patterns of SSWs and PWDs from ERA5, which are generally similar to but noisier than that from the model. Figs. A1 and A2 show additional maps like Fig. 8 but for 2m temperature and precipitation, respectively.

## 4 Summary and conclusion

The results from this paper challenge the general belief that the reversal of the polar vortex associated with major sudden stratospheric warming events (SSWs) is the key physical element for the creation of stratospheric signals at the surface. Building upon earlier work by Polvani and Waugh (2004), and using a long control run with a stratosphere-resolving coupled climate model, we showed that the accumulated upward-directed lower stratospheric wave activity flux is a more effective indicator for major polar vortex perturbations than SSWs. We used the wave activity flux to define so-called positive wave driving events (PWDs), which by construction had the same occurrence frequency as SSWs. Much of the study was then about understanding the similarities and differences between PWDs and SSWs. About half of all PWDs occurred at the same time as SSWs, and just like SSWs, PWDs were followed by abrupt decelerations of the polar vortex and long-lasting negative anomalies of the North Atlantic Oscillation index at the surface.

However, half of all PWDs did not concur with major SSWs, pointing to important differences between PWDs and SSWs. For example, since our definition of PWDs also permits dynamical final warmings and SSW-like events in April, PWDs were more evenly distributed over the winter than SSWs. There was also the indication that PWDs are more sensitive to the influences from ENSO than SSWs. Perhaps most importantly, PWDs tended to be followed by stronger surface responses than SSWs, and this had two principal reasons. First, the PWD definition excluded many of the weak SSWs in late winter, associated with relatively small wave activity fluxes and surface responses. Second, the PWD definition included mid-winter polar vortex events, which formally did not fulfill the SSW definition, but which were associated with strong wave activity fluxes and robust surface signals. Apparently, the reversal of the polar vortex is a less important criterion for creating downward propagating signals than the strength of the wave activity flux and the relative perturbation of the polar vortex. This interpretation is consistent with earlier findings that strong upward wave activity fluxes are a good indicator for a downward propagating response of SSWs (Karpechko et al., 2017; White et al., 2019).

Besides being a valuable measure for stratosphere-troposphere coupling and identifying events with a robust surface impact, there are more advantages to the PWD definition. For example, PWDs

- capture with one criterion a variety of event types, including major warmings, minor warmings, final warmings (FWs), and also strong vortex events;
- detect strong SSWs and also dynamical FWs, but avoid weak events that have little surface impact; and
- may lengthen the forecast horizon for polar vortex events because the wave activity flux precedes the onset of events.

On the other hand, there are also disadvantages to the PWD definition. First, it requires knowledge of EP-fluxes, which are more complicated to calculate than the simple zonal mean zonal wind for SSWs. In addition, EP-fluxes are often unavailable from models, highlighting the need to make these and other dynamical variables publicly available to projects like DynVarMIP (Gerber and Manzini, 2016). Lastly, for certain applications, it may be a disadvantage that PWDs do not distinguish between different event types. However, some of this information can be easily added, as done in this study for U- and U+ events.

We also considered the ERA5 reanalysis in this study. Despite the limitation from the small number of observed events, we found similar outcomes from the ERA5 as for the model. We also underline that our model has a quite realistic circulation, which gives us confidence that our model results are indeed applicable to the real world.

For a better comparison with SSWs, we used a fixed PWD wave driving threshold. However, for practical applications there is no need for specific thresholds. On the contrary, every event is associated with a different amount of wave driving, allowing for a spectrum of events with different magnitudes, similar to the classification of other extreme events like hurricanes or tornadoes. To illustrate this, we calculated from the ERA5 the occurrence of PWDs and NWDs with an absolute accumulated wave driving of at least 5 WDU·day. As shown in Fig. 9, at this reduced threshold, there occurred several wave driving events per year of either sign, and the events are followed by consistent perturbations of the polar vortex winds. These perturbations are not always strong, but they may still create useful surface signals for S2S predictions.

While this study was only concerned with vortex events over the Arctic, the wave driving definition can also be used to detect vortex events over the Antarctic. There, SSWs are extremely rare and seasonally phase-locked towards the end of winter (Jucker et al., 2021; Wang et al., 2020), creating the need for a more practical definition with more frequent events. The problem is that the polar vortex over the Southern Hemisphere (SH) is stronger than over the NH, making it more difficult for the SH vortex to reverse its direction despite the occasional occurrence of strong wave forcings and vortex responses. The research community has already started to test some alternate definitions for SH vortex events (Thompson et al., 2005; Lim et al., 2019; Jucker et al., 2021), and it is up to future research to compare these and other definitions against the wave driving approach taken in the present study.

## 5 Appendix

## 6 Author contribution

TR designed the study, performed the model simulation, analysed the data, and wrote the manuscript. MJ was involved in the scientific interpretation of the results, the design of the research questions, and the review of the manuscript.

## 7 Competing interests

The authors declare that they have no conflict of interest.

## 8 Acknowledgements

We thank the U.S. National Science Foundation (NSF) and the Australian Research Council (ARC) for their support. We thank ECMWF and the Research Data Archive at NCAR/CISL for providing the ERA5 data. The coupled model simulation for this study was a major effort and required several years to run at the Center for High Performance Computing (CHPC) of the University of Utah. We gratefully acknowledge CHPC for its support. We thank Darryn Waugh for his helpful comments on an earlier version of this manuscript. We also thank Amy Butler and another anonymous reviewer for their thoughtful comments.

## 9 Financial Support

TR was supported by NSF under award number 1446292 and 2103013. MJ was supported by ARC under grant FL150100035.

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

**Figures, Figure Captions, and Tables**

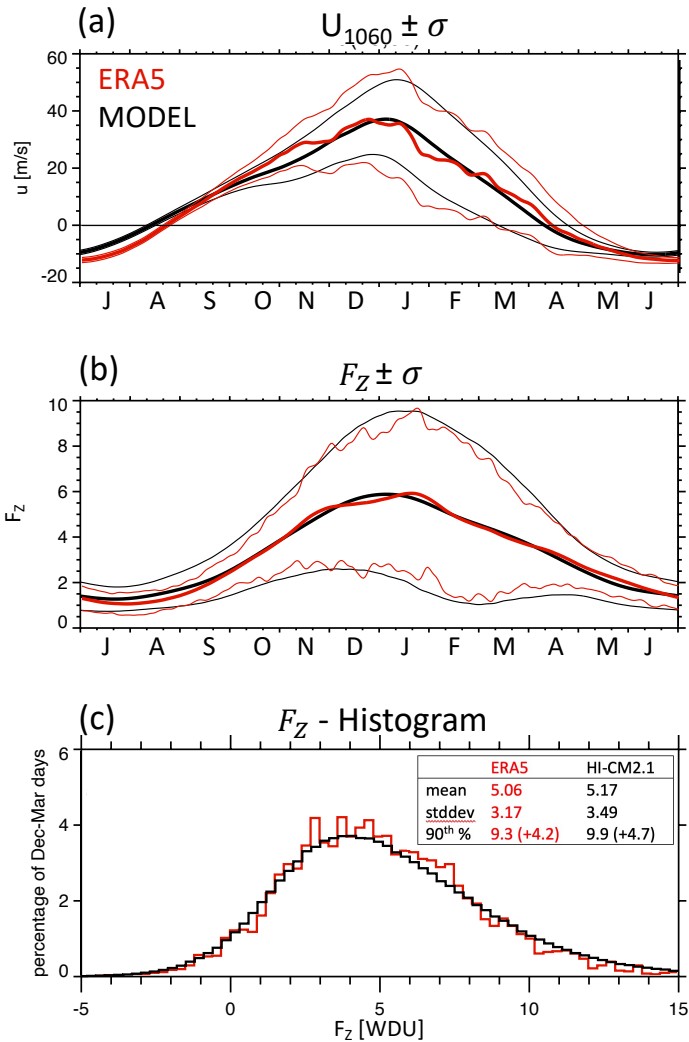

Figure 1: Model validation against ERA5 reanalysis (1979-2020) for (a) the (thick) climatological seasonal cycle of the zonal-mean wind at 10 hPa and 60°N $u_{1060}$ (m·s$^{-1}$) and (thin) its ±1 standard deviation, (b) the (thick) climatological seasonal cycle of the stratospheric wave driving $F_Z$ (-10$^5$ kg·m·s$^{-4}$, or WDU) and (thin) its ±1 standard deviation, and (c) the daily distribution of $F_Z$ during northern winter (DJFM). Inset in (c) shows the mean, the standard deviation, and the 90-percentile of the two distributions.

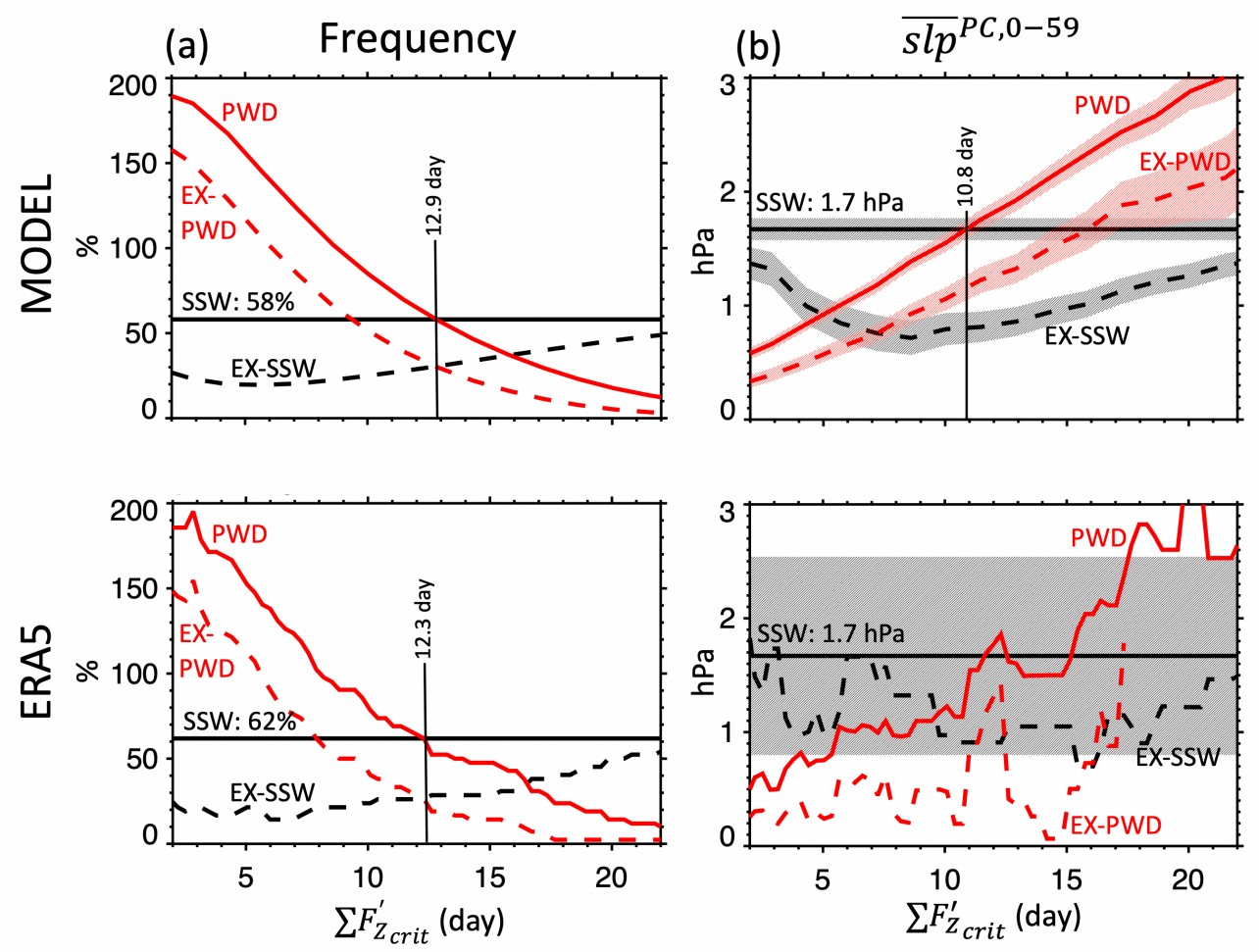

**Figure 2: Sensitivity of (a) event frequency (%) and (b) surface response (hPa) to the wave driving $\sum F'_{Z_{crit}}$. Event frequency is given in %, i.e., events per year multiplied by 100. The surface response in (b) is the polar-cap-averaged (60-90°N) sea level pressure (SLP) anomaly averaged from day 0 to 59 after onset ($\overline{slp}^{PC,0-59}$). Thick horizontal line is for SSWs. Shading in (b) shows the 95% confidence intervals. EX-PWDs and EX-SSWs are events that are either only PWDs or only SSWs. The top panels for the model show that (a) at $\sum F'_{Z_{crit}}$ = 12.9 day the frequency of PWDs is similar to SSWs, and (b) that at $\sum F'_{Z_{crit}}$ = 10.8 day the SLP response to PWDs is similar to SSWs.**

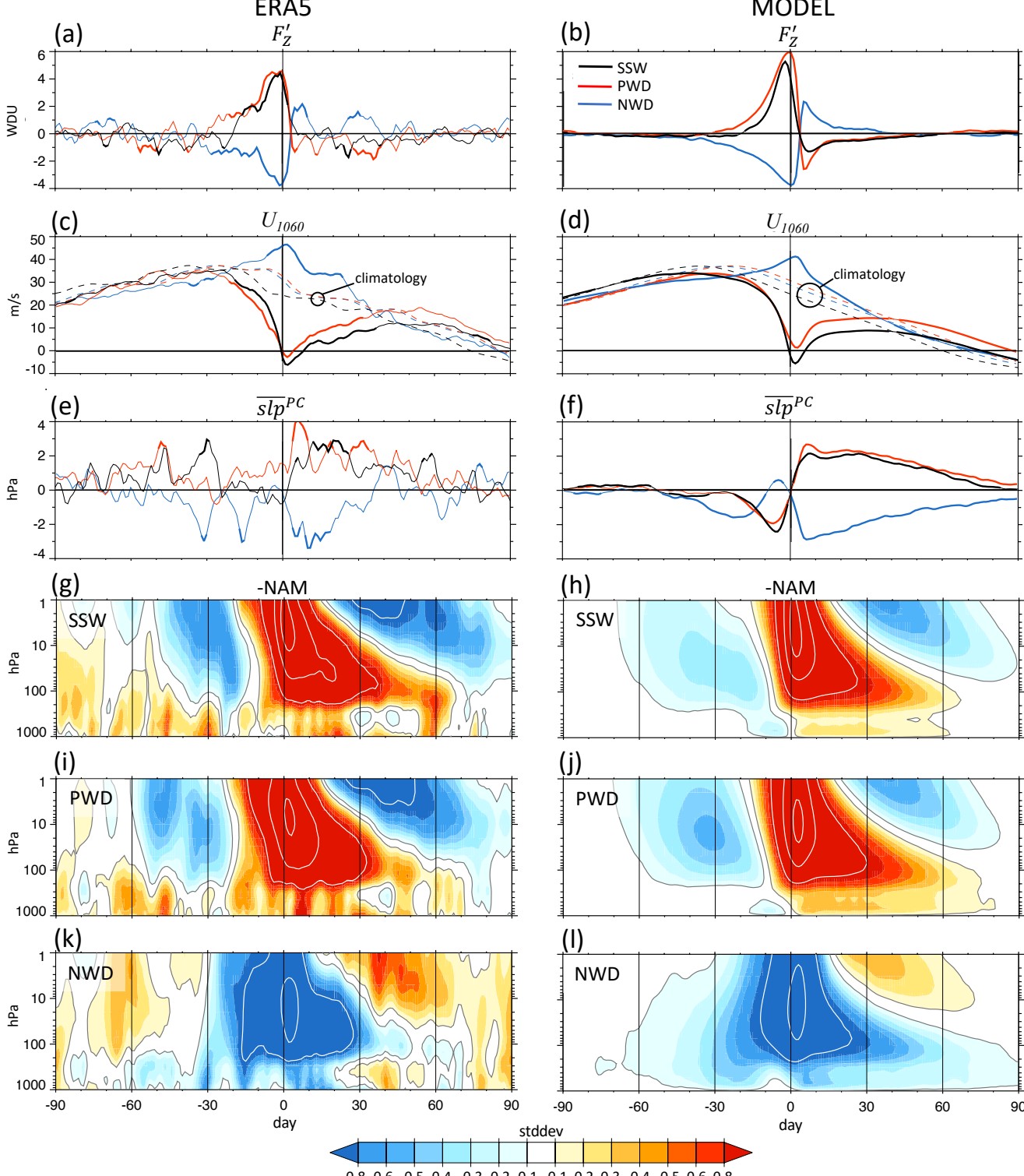

**Figure 3: Composite-mean lifecycle of SSWs, PWDs, and NWDs.** Shown are (a-b) anomalous wave driving $F_z'$ (WDU), (c-d) $u_{1060}$ (m/s), (e-f) anomalies of the polar cap averaged (60-90°N) SLP ($\overline{slp}^{PC}$) (hPa), and (g-l) time-height cross sections of standardized polar-cap-averaged geopotential height anomalies (stddev); positive (negative) anomalies correspond to negative (positive) NAM anomalies and are shaded reddish (blueish) from ±0.1 to ±0.8 standard deviations; additional white contours are shown at ±1, ±2 and ±3 standard deviations. Bold lines in (a-f) show anomalies that are significant according to a two-sided Student *t*-test at the 5% error level. Dashed thin lines in (c-d) show the associated climatological $u_{1060}$, aligned with the onset dates of the respective events. Left panels are for the ERA5 (26 SSWs and 26 PWDs; 62%), and right panels are for the model (5225 SSWs, 5226 PWDs, and 4068 NWDs). The wave driving threshold $\sum F'_{Z_{crit}}$ is 12.3 day for ERA5 and 12.9 day for the model.

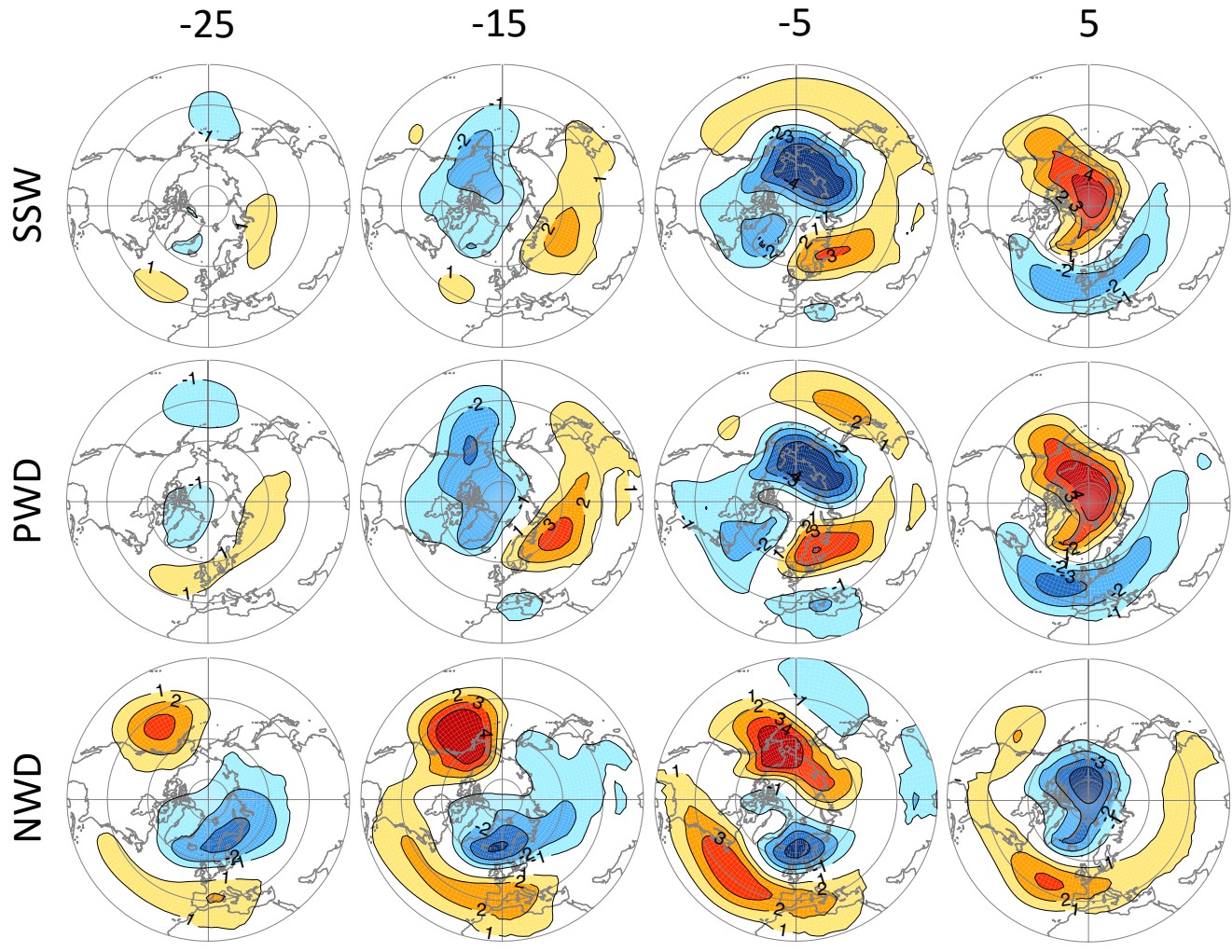

**Figure 4: SLP precursors for model-generated (top) SSWs, (middle) PWDs, and (bottom) NWDs during neutral ENSO years. Shown are composite 10-day-mean SLP anomalies (hPa), centered on the lag days shown on top. Contour interval is 1 hPa. Reddish and blueish shadings denote positive and negative anomalies, respectively, which are all statistically significant at the $p <$ 0.05 level. The number of SSWs, PWDs, and NWDs is 2607, 2291, and 2172, respectively. The wave driving threshold $\sum F'_{z_{crit}}$ is 12.9 days. The composite Nino 3.4 index for SSWs, PWDs, and NWDs is -0.09 K, 0.04 K, and -0.24 K, respectively**

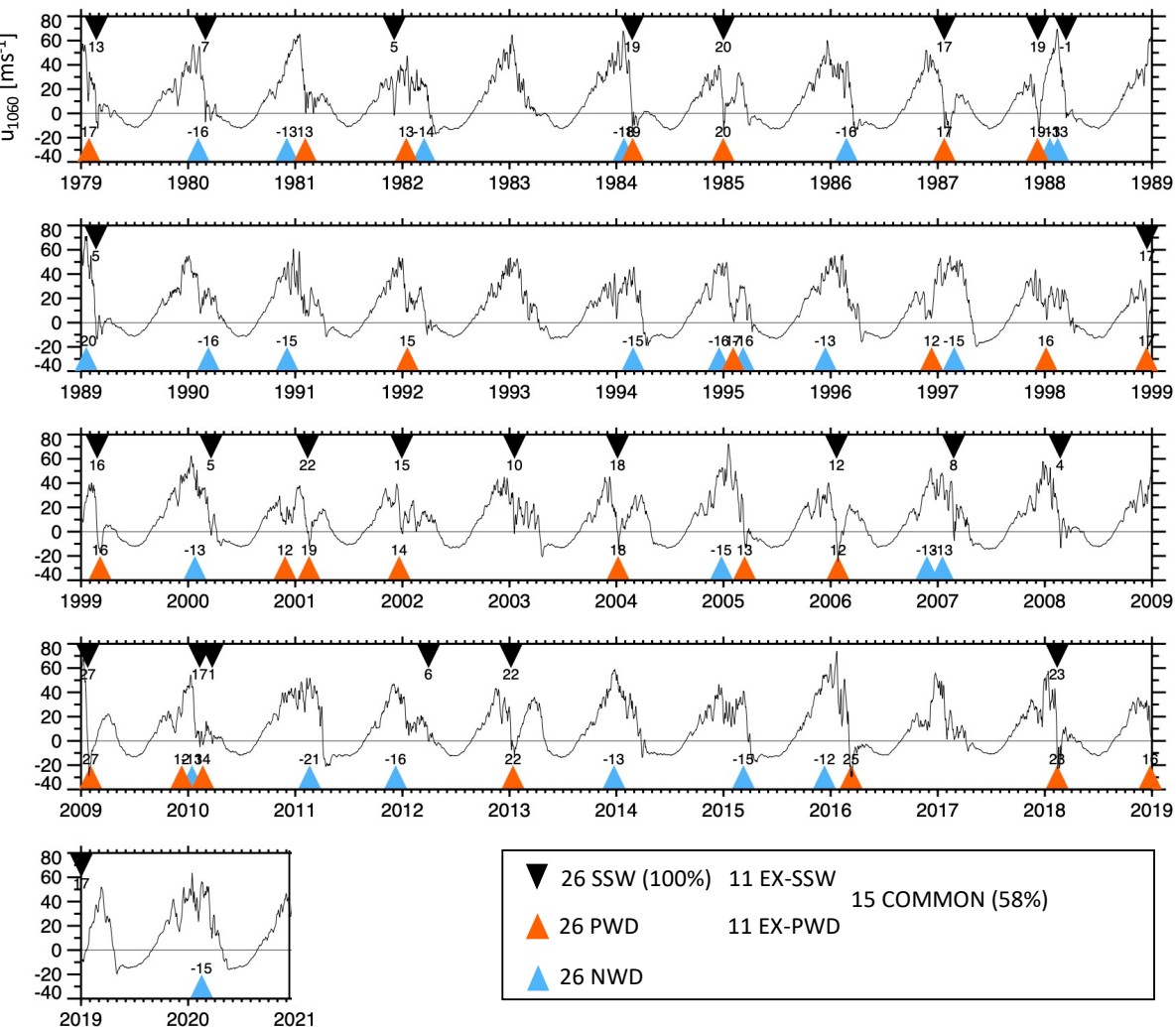

**Figure 5:** $u_{1060}$ time series (m/s) and history of SSWs, PWDs, and NWDs in ERA5. Horizontal axis shows years; triangles show event day of year (0 = 1/1), small numbers indicate $\sum F'_Z$ in days. The PWD wave driving threshold $\sum F'_{Z_{crit}}$ is 12.3 days.

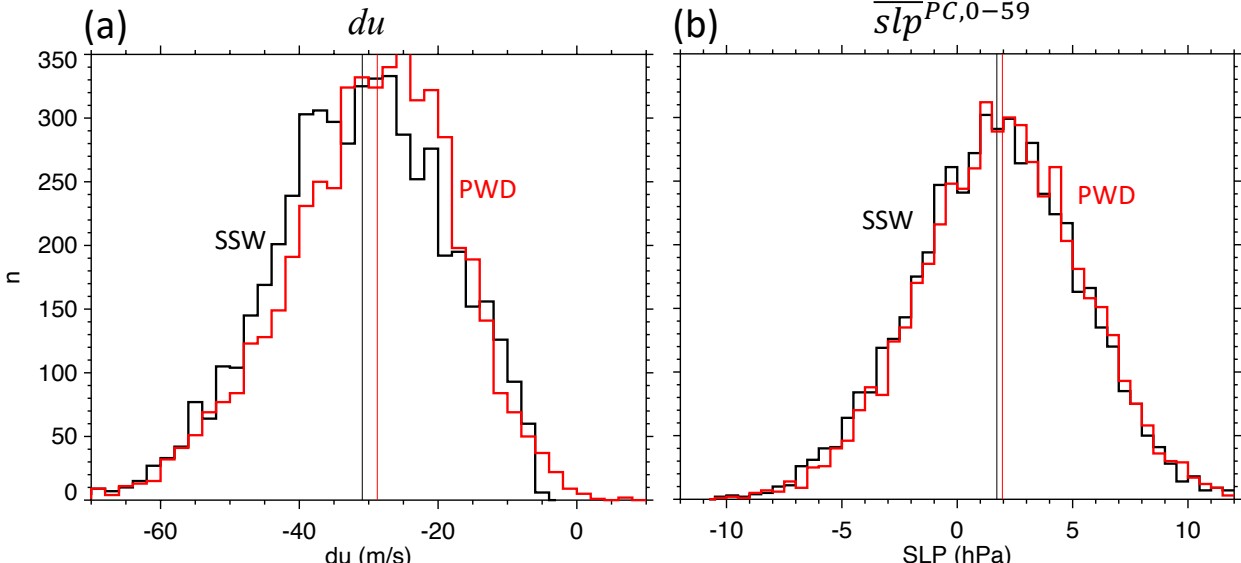

**Figure 6: Response histograms of model-simulated SSWs and PWDs in stratosphere and troposphere. Shown are responses to all (black) SSWs and (red) PWDs in terms of (a) the minimum anomalous $u_{1060}$, $du_{min}$, and (b) the polar cap averaged SLP following the events ($\overline{slp}^{PC,0-59}$). Vertical axis gives number of events $n$ per bin. Thin vertical lines give the medians of the distributions. PWD wave driving threshold $\sum F'_{z_{crit}}$ is 12.9 days. Shown are the outcomes from 5224 SSWs and 5154 PWDs.**

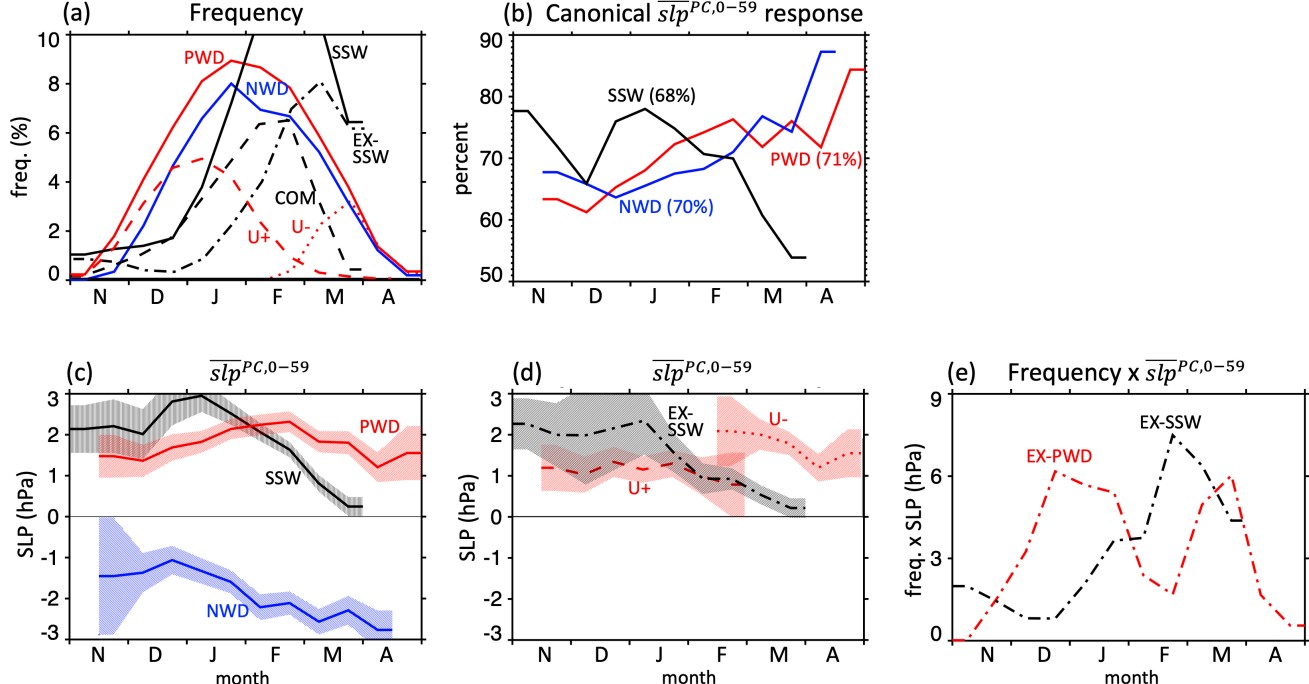

**Figure 7: Seasonality of event frequency and surface response in the model. Shown are histograms of (a) the frequency of events (%), (b) the percentage of cases that $\overline{slp}^{PC,0-59}$ has the expected sign (> 0 for SSWs and PWDs; < 0 for NWDs), (c-d) $\overline{slp}^{PC,0-59}$, and (e) the product of frequency and $\overline{slp}^{PC,0-59}$ averaged over all cases within one bin for the shown events as a function of the day of the year (horizontal axis, Nov. – Apr.). PWD wave driving threshold $\sum F'_{z_{crit}}$ is 12.9 days, and bin-size is 15 days. Partly hidden black curve for SSWs in (a) peaks at the end of February at ~13%. Shadings in (c) and (d) show the 95% confidence intervals determined from a t-test. Black is for SSWs, red is for PWDs, and blue is for PWDs. Continuous lines show all events of one class (SSWs, PWDs, or NWDs), dashed red lines in (a) and (d) are EX-PWD U+ events, dotted red lines in (a) and (d) are EX-PWD U- events, black dashed lines in (a) are common events (COM), and dashed-dotted lines in (a) and (e) are EX-PWD (both U+ and U-) or EX-SSW events.**

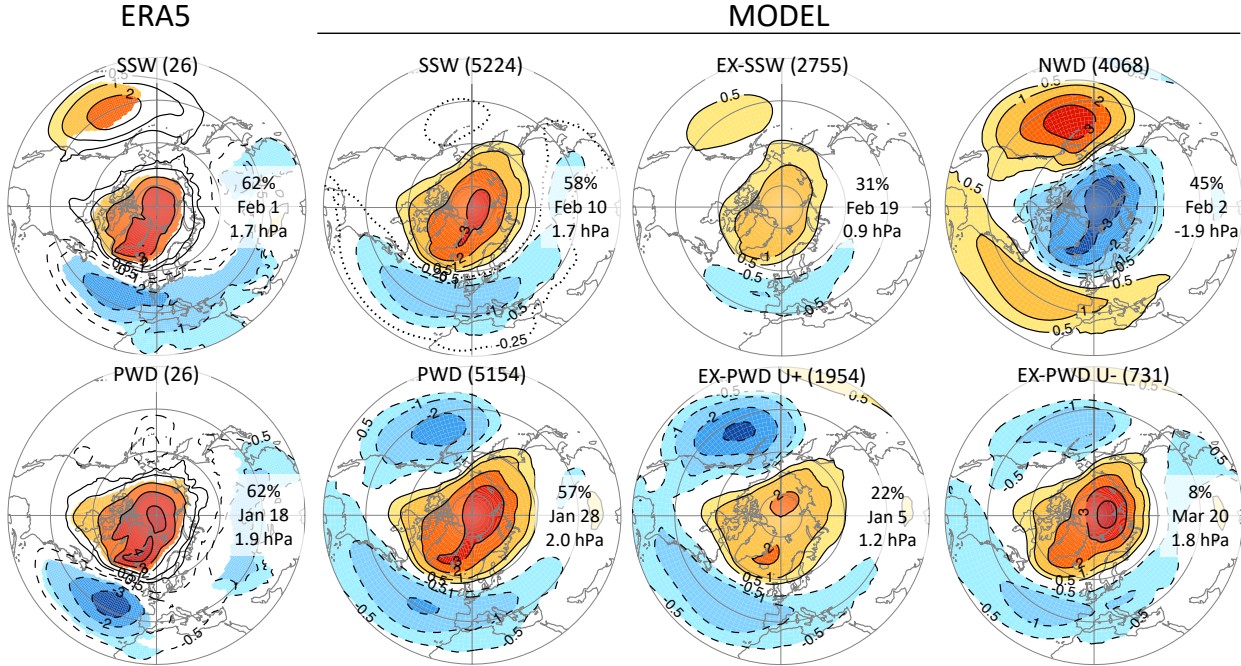

**Figure 8: SLP response to stratospheric events.** Shown are composite SLP anomalies (hPa), averaged from day 0 to 59 after event onset. (left) column is for ERA5, remaining columns are for the model. Shadings indicate anomalies that are statistically significant according to a two-sided t-test at the 5% error level. Contours are at ±0.5, ±1, ±2, ±3, ±4 hPa; extra dotted contour at -0.25 hPa is only shown for model simulated SSWs. Annotations in each map indicate (i) absolute frequency, (ii) mean onset date, and (iii)

$\overline{slp}^{PC,0-59}$. PWD wave driving threshold $\sum F'_{Z_{crit}}$ is 12.9 days for the model and 12.3 days for ERA5. EX-PWD are PWDs that are not SSWs, and they are shown separately for U+ ($u_{min} > 0$) and U- ($u_{min} < 0$).

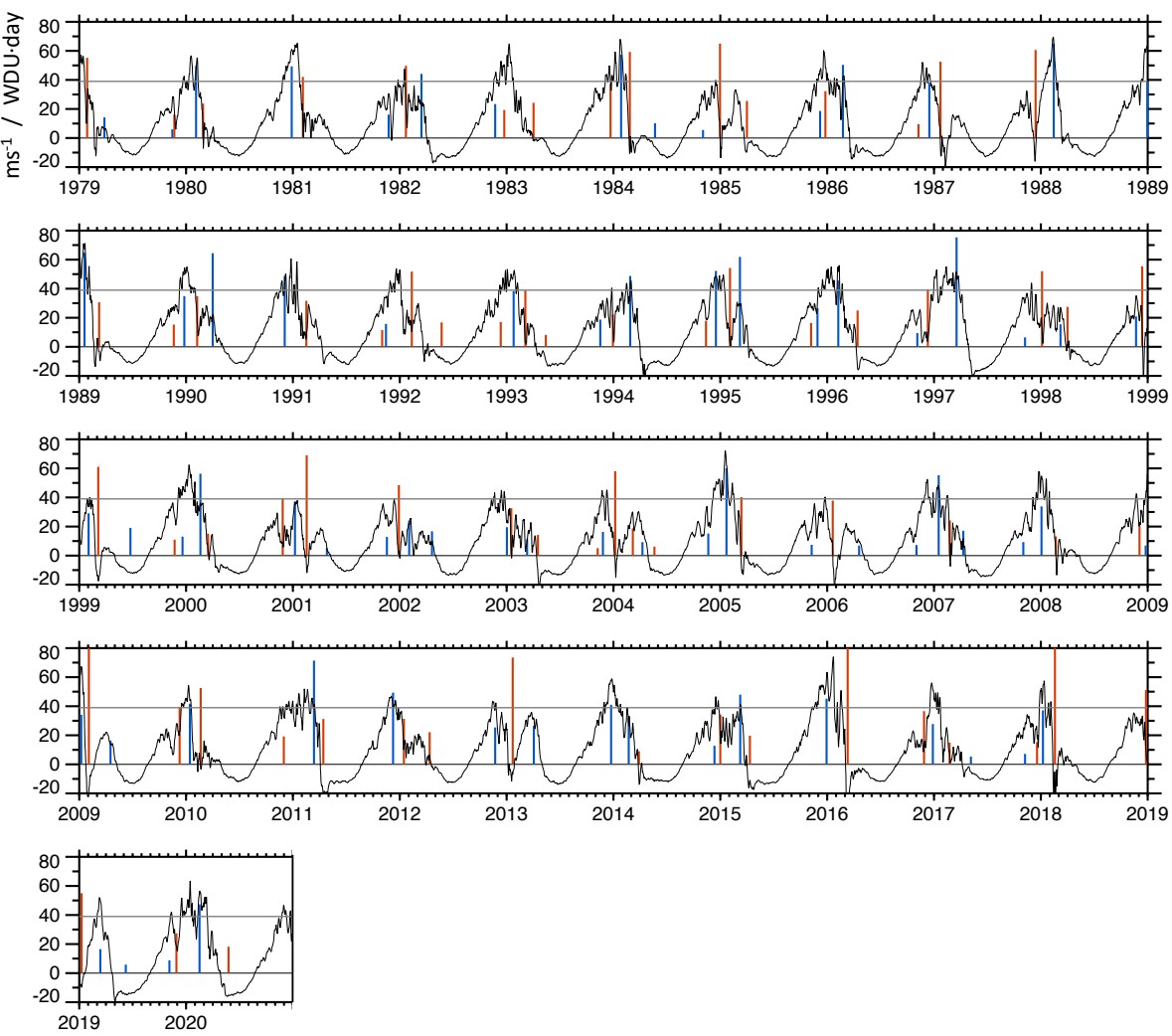

**Figure 9: ERA5 wave driving events with $|\sum F'_z| \geq 5$ WDU·day. Vertical bars show onset dates of individual PWD (red) and NWD (blue) events, with length of bars indicating $|\sum F'_z|$ of the events (in WDU·day). Events are only shown if there is no other same-signed stronger event nearby (< 60 days). Black curve shows the corresponding $u_{1060}$ time series (m/s) and horizontal axis shows years.**

**Table 1: Statistics of the accumulated wave driving anomaly, $\sum F_Z'$. Tabulated are the mean, maximum, and minimum of $\sum F_Z'$ (units: day) prior to the onset of the various events. The threshold $\sum F_{Z_{crit}}'$ for PWD and NWDs is ±12.9 days for the model and**

 **±12.3 days for the ERA5.**

| | ERA5 | | | Model | | |
|---|---|---|---|---|---|---|
| | SSW | PWD | NWD | SSW | PWD | NWD |
| mean | 13.2 | 16.9 | -14.9 | 12.5 | 17.5 | -13.2 |
| max | 27.0 | 27.0 | -12.3 | 42.5 | 42.5 | -12.9 |
| min | -0.7 | 12.3 | -20.6 | -18.5 | 12.9 | -34.9 |

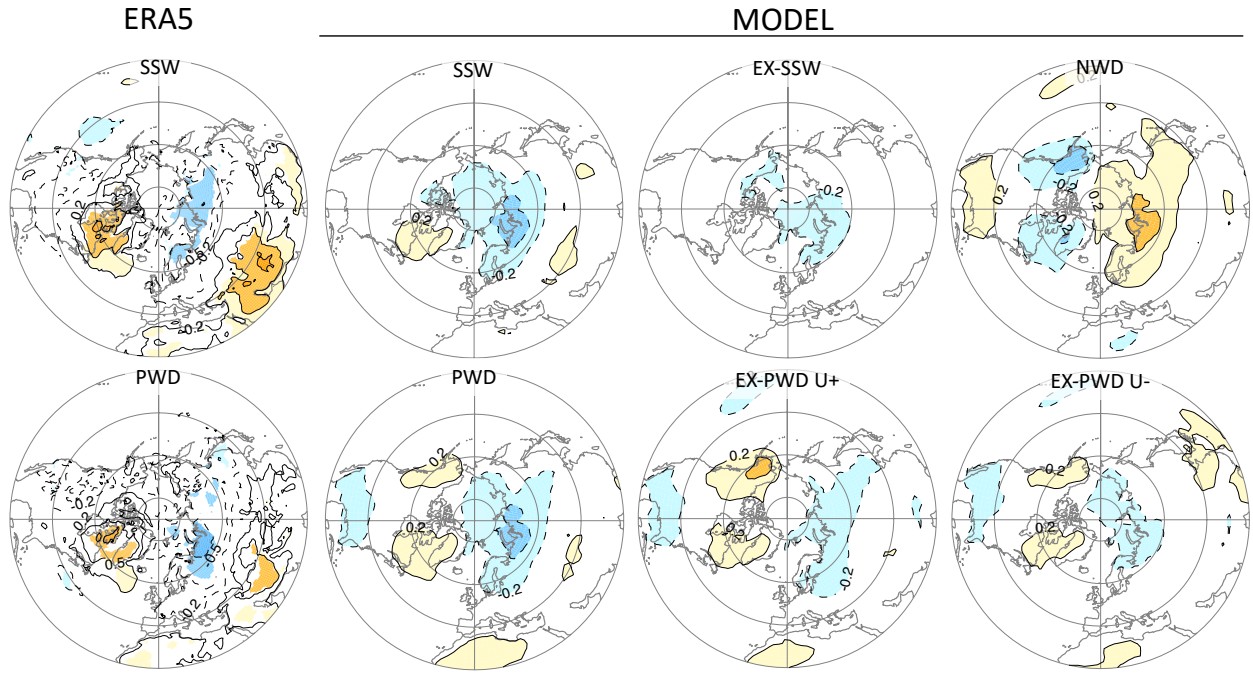

**Figure A1: As Fig. 8 but for 2m temperature. Contours are at ±0.2, ±0.5, ±1, and ±1.5 K.**

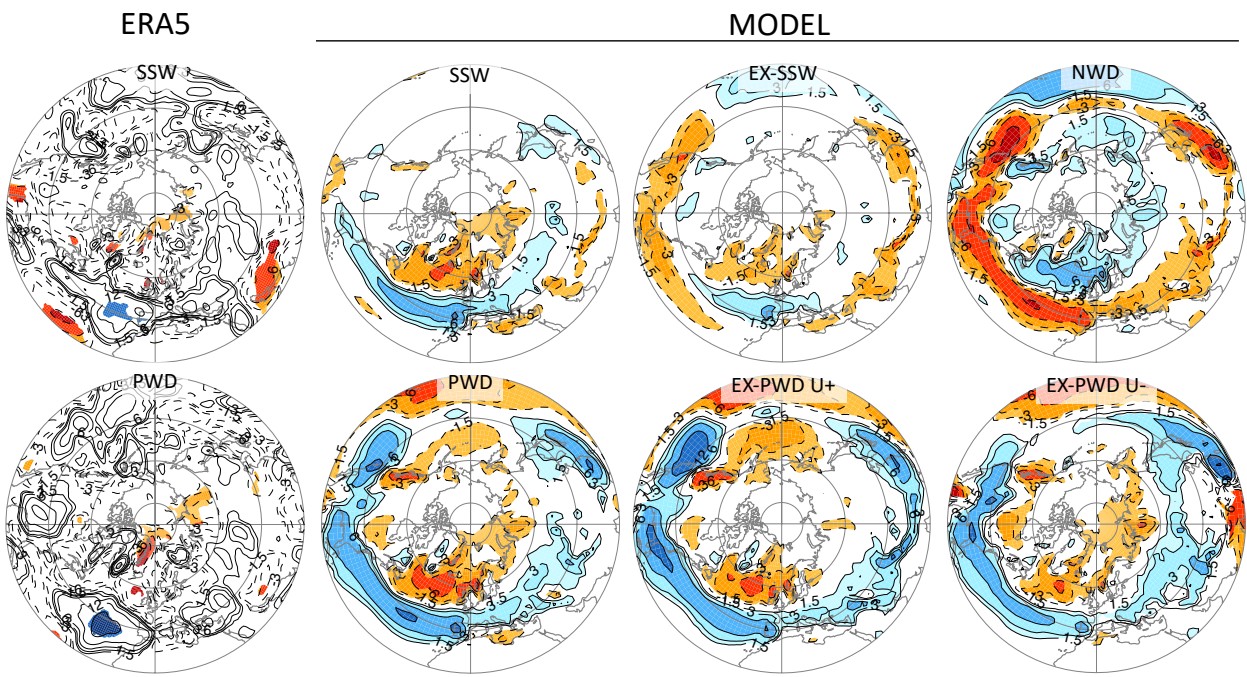

**Figure A2: As Fig. 8 but for precipitation. Contours are at ±1.5, ±3, ±6, ±12, ±24 mm/month.**