# Peer review of "Stratospheric wave driving events as an alternative to sudden stratospheric warmings"

_Weather and Climate Dynamics, 2022_

## Author Response (AR1)

**RC1**

**General Comments**

Overall, I enjoyed reading this manuscript and the results were interesting and novel. I mostly have minor comments, with a few more broad suggestions for improvement.

First, while negative wave driving events (NWDs) are included in the study, they seem almost an after-thought. It would be nice to include something on NWDs in Figures 3 and 4 for example (or in a simple manner to Figures 3 and 4 but in an appendix). More discussion or even a separate section could be given to discussing these events. Or, another option would be to consider not including NWDs here but save these for another study (perhaps comparing to "vortex intensification" events instead of SSWs).

We modified Figs. 3 and 4 to also include NWD events. We also provide some additional discussion of the results for NWDs in the text.

Second, I had quite a few comments on the Discussion/conclusions section below, that I hope the authors will consider also more broadly for the entire manuscript. While I find the results presented in the study very compelling, I also think that the primary goal of the study should not be to dismiss the major SSW definition. In particular, some of the text seems to imply that PWDs and SSWs are different phenomenon, while they really are just two different ways of measuring the same basic dynamical process. While I agree with the authors that the wave driving events do have advantages over SSWs particularly if the interest is in surface impacts, rather than trying to argue that SSWs are not important or the SSW definition not valid, I think the results of the study could be better used to argue for improved dynamical metrics output from forecasting/climate models in order to encourage use of this somewhat more complicated metric (and, I do think it worth acknowledging that while the wave activity metric is relatively straightforward to calculate, it does require much more data than u1060).

We agree and try to modify our language throughout the entire document. For example, we now add to the conclusion section a discussion of some of the disadvantages of the PWD definition.

**Specific Comments**

Title: "sudden" appears twice. More generally, I suggest in the title and in the text (e.g., line 11, line 368) that the acronym for SSW is defined as "sudden stratospheric warming" rather than "stratospheric sudden warming"; reasons for which are described in Butler et al. 2015, footnote 1.

We made the proposed change(s). But it is interesting, going through our list of references, about half of the paper titles use one version (sudden stratospheric), and the other half use the other version (stratospheric sudden). It seems, we all get sometimes confused about this issue.

Line 12, line 35: "normal westerly"- to be clearer, specify "climatological wintertime westerly"

Corrected

Line 13-14: Since the event date itself is presumably determined by the wave activity flux, remove "prior to the onset of the events".
Corrected

Line 15-18: phrasing is confusing, plus this point is repeated in the next line. Suggest instead just stating "About half of the wave events are identical to SSWs. However, there exist several advantages for defining stratospheric weak extremes based on wave events rather than using the common SSW definition:…"
Corrected

Line 31: What is meant by "inner core"? In the NH the warming often seems to start in one particular extratropical sector and move inwards towards the polar cap.
Inner core refers to the region poleward of the wind maximum at ~60N. As you say, the warming moves inward, so, I think "inner core" expresses this quite nicely. If you disagree, could you perhaps suggest a better wording?

Line 41: Could you be more specific in terms of latitude/level of the CP07 definition?
Corrected

Line 48: Here, I think you mean "it is unclear how effective this definition of SSWs is in capturing events with downward influence…"
Corrected

Line 51-52: If the zonal wind reverses, how can it be considered a relatively minor perturbation? According to your prior paragraphs, the reversal is why major SSWs are considered so intense. Since this sentence also repeats the sentiment of the prior one about downward impacts, I would just remove it. The point about it being a based on a fixed threshold is perhaps really only important in the context of climatological model biases; or regarding your next statement on lines 53-54, which is a valid one.
Maybe we did not explain this very well. We now write hopefully more clear: "Another downside of the CP07 definition is that it is based on a fixed threshold and, as long as the zonal wind reverses, the definition also detects the perturbation of a climatologically weak vortex that presumably has a small surface impact."

Line 55-57: Is this your wave driving definition applied here to the SH? If not, it seems somewhat out of place to bring this up here (I like the point about the SH in the conclusions, but I would remove it here and focus only on the NH).
We removed the last two sentences and changed some of the SH-SSW discussion in the conclusion.

Line 17, line 58, line 125, 405: Not sure about the word "traditional". Technically the traditional WMO one is not exactly the same as the CP07 definition; see Butler et al. 2015, which describes

the long history of SSW definitions. Perhaps you should instead use the word "common" to describe this definition and state somewhere that by "common SSW definition" you mean the Charlton and Polvani definition.

We eliminated the words "traditional definition" from the entire document and replaced them by "CP07 definition".

Line 60: This paper should be cited here or elsewhere in the manuscript; it is very relevant: https://journals.ametsoc.org/view/journals/clim/30/14/jcli-d-16-0465.1.xml

We now cite this paper, thanks for pointing this out.

Line 60-62: I agree, the SSW definition is not based on what precedes the event, but on the other hand is also not based on "their effect", at least at the surface, which is how this may be interpreted. Maybe "but instead on their stratospheric effect". I would also argue that a wind reversal is very much a "dynamically-oriented" metric, so I would rephrase (see also, Butler and Gerber 2018, which showed that the particular latitude and height of the CP07 definition does maximize dynamic metrics, at least for definitions based on wind reversals).

We changed "effect" into "stratospheric effect" and removed the "more dynamically oriented alternative".

Line 61: This is the first time the predictability of SSWs is brought up. Could you make a brief statement about what lead times are expected for SSW prediction? (e.g., Domeisen et al. 2020 part I).

We now write: "This may be relevant for the prediction of polar vortex events in the context of S2S applications, since a more cause-centered approach could lengthen the relatively short 1-2 week-long predictability limit for polar vortex events (Domeisen et al., 2020c)"

Line 74-96: There are other studies that suggest caution about how to interpret the 100 hPa eddy heat flux, which should perhaps be mentioned in this section somewhere, see, e.g.: https://journals.ametsoc.org/view/journals/atsc/74/9/jas-d-17-0136.1.xml

Thanks for pointing this out. We now added into the manuscript: "We note that the flux at 100 hPa should not be simply interpreted as a wave propagation from the troposphere into the stratosphere. Cámara et al. (2017) showed that only 1/3 of the wave flux variance at 100 hPa can be explained from the flux in the upper troposphere (300 hPa). They argued that the 100 hPa level is well above the extratropical tropopause and is thus already under the considerable influence of stratospheric processes. In the context of our study, however, the exact source for the wave activity flux is less important."

Line 86: Do you test different levels? What about the wave activity at 50 or 10 hPa, for example? 100 hPa could still be muddied by tropospheric variability potentially.

No, we did not test other levels. In choosing the 100 level we simply followed previous work. Testing the sensitivity to other levels would require lots of extra work and almost seems the subject of a new study. Also, we feel that the level should be as low as possible as to maximize the propagation time of the waves into the middle stratosphere.

Line 121: what is the model lid height?
We now added: "… , a model lid at 0.002 hPa (ca. 92 km), …"

Line 125: The CP07 definition is not exactly the WMO-criterion. See Butler et al. 2015.
We now avoid the reference to the WMO and use instead everywhere "CP07 definition".

Line 129, line 151: this should read "separated by at least 20 consecutive days of westerlies".
See CP07 corrigendum: https://journals.ametsoc.org/view/journals/clim/24/22/jcli-d-11-00348.1.xml. In your methodology described on line 151, are your events just separated by 20 days (regardless of wind or PWD sign?). Please clarify.
Good catch! We made the corresponding change. Since our algorithm is determined by (the quite noisy) FZ100, we only ask that two events must be separated by 20 or more days. We now write more precisely: "As for SSWs, multiple PWDs in the same winter must be separated by 20 or more days, but there is no extra requirement for the sign of $F_z'$ or $u_{1060}$ during this period."

Line 134: This seems like a broad latitude range to calculate Fz. Did you also try 45-75N or 40-80N as many other studies use?
No, we did not. Are you concerned about the heat flux calculation reaching too much into the tropics? And, are we the only ones who use this wider range? I really don't recall the exact reasons, but we made this decision a long time ago. From my experience, Fz peaks at ~60N, and as long this latitude is included, the exact latitude range does not matter much. Also, I believe, the contributions from 20-45N are small. From Polvani & Waugh (2004), who indeed used 45-75N: "Similar results are obtained if the vertical component of the E–P flux is used or if averaging occurs over a wider latitude range."

Lines 134-136: I don't follow why multiplying by -1 ensures that a positive sign means upward propagation. Which term in Fp leads to upward propagation being negative?
Fp is negative for poleward heat flux/upward wave activity because of the partial_theta/partial_p term.

Line 141: "of either sign"- this is confusing because the paragraph starts off describing "positive wave driving events" yet here you are looking for anomalies of Fz of either sign.
We added "of either sign" to underline that the daily Fz values can of either sign. Only the integrated Fz must be positive for a positive wave driving event to be detected. We now write hopefully more clearly: "daily $F_z'$ values can be of either sign".

Line 150: What does it mean for this quantity to become negative? It's not clear in a physical sense how that signals a positive WD event. Perhaps an example would be useful? (it does become a bit more apparent in Figure 3).
When the daily Fz turns negative it indicates the end of the preceding wave burst. This determines $t_0$, and we then continue the search for another PWD event. We now write: "We further advance $t_0$ until the daily $F_z'(t_0)$ becomes negative. This determines the end of the wave event and the final onset date $t_0$ of the PWD."

Line 157: du_min is not mentioned on line 129-130 as being saved for SSWs; might do so for consistency. Also the statistics listed here don't match what is listed in Table 1.

We corrected line 129-130 by also mentioning du_min. We also corrected out wording: "Unsurprisingly, in the mean, SSWs and also PWDs have positive $\sum F_Z'$ values, …".

Line 199: This isn't apparent from the figure (the EX-PWD line falls beneath the SSW value at the 12.9 threshold).

The EX-PWD line falls below the SSW value at ~16 WDU (not at 12.9), which means that PWDs with a wave driving of >16 WDU (therefore we used the word "some") create on average a stronger surface response than SSWs. We now write: "We also note that strong EX-PWDs, i.e., polar vortex events with $\sum F_Z' > 16$ WDU but which are missed by the SSW definition, create on average a stronger SLP response than SSWs".

Figure 2: This is a really nice figure. What are the units of event frequency (is this number of events per year multiplied by 100?)

Yes, correct. To clarify this, we now write at the beginning of this section: "Here and everywhere else, the event frequency is given in %, i.e., events per year multiplied by 100."

Line 202: It would be nice to at least briefly mention that especially the SLP response in ERA5 is much noisier due to the smaller sample size, which means that the SLP difference between PWD and SSWs and the selected threshold is not significant in ERA5. I think the overall good agreement with the model does support the idea that using ERA5 here alone would be difficult and provides support for needing many samples from model experiments.

We now add: "But due to the smaller sample size, the results are much noisier and the SLP difference between PDWs and SSWs is not significant."

Line 220: I don't know if it will be clear to the general reader what is meant by the final warming here (I don't think it's been defined yet) and which zero-wind crossing you are referring to.

We now add: "Final warmings (FWs) represent the last weakening of the polar vortex at the end of winter when the seasonal increase of radiative heating over the pole prevents the reformation of the vortex (Black and Mcdaniel, 2007)." We also add: "80-90 days after the onset of SSWs and PWDs".

Line 229: I can see the "eastern" vs "western" dipole in the lag -15 panel but not so much in the lag -25 panel.

True, but some remnants of the dipole can still be seen at lag -25, in particular for PWDs.

Line 248-253: Here, you have confused the sign of the NAM. The positive values here represent negative values of the NAM (at least, I hope this is the case!). The text should be fixed here; and since the NAM here is approximated by the polar cap geopotential height anomalies, either these should be labeled as such on the figure (instead of as the NAM), or they should be multiplied by -1 to match NAM polarity conventions.

Oh God, how embarrassing! We now fixed the text and added: "Note that negative (positive) NAM anomalies correspond to positive (negative) polar-cap-averaged geopotential height anomalies and are therefore shown in Fig. 3 by reddish (blueish) shading". We prefer to leave Figs. 3g-j as they are to better match the original work by Baldwin and Dunkerton and added minus signs in front of the NAM labels.

Figure 5: Instead of event day of year, which doesn't seem that useful since it can be somewhat inferred from the location of the triangle, it would be more useful to put the accumulated wave driving value for each event as the small number.
We changed Figure 5 to now show the accumulated wave driving. We also provide some new discussion in the text.

Line 271: "when the polar vortex is weak" – would instead phrase as "when the polar vortex is climatologically weaker"
Corrected.

Line 283-286: I understand your point here, but on the other hand, each SSW also evolves differently, in terms of the morphology of the vortex, so in a similar fashion there is also much uncertainty about how that contributes to the surface response and timing of the response in terms of individual cold air outbreaks. This is of course underscored by your Figure 6 and the following discussion.
Yes, the main point is that strong upward wave activity flux does not guarantee a strong vortex response, adding to the already existing uncertainty.

Figure 7: It's hard to tell whether the SSW curve is flat at the top or extends above the y-axis. It may be preferable to extend the y-axis so the peak can be seen.
The curve for SSW is not flat at the top (see Figure 1 below), but we prefer to not change the y-axis of Fig. 7a. Otherwise, there would be not enough space for the many curves and their labeling. We now explain in the caption: "Partly hidden black curve for SSWs in (a) peaks at the end of February at ~13%."

Also for panel (d); it's hard to see what is going on because the EX-SSW blocks the U+.
Also, there is a technical problem of how figures with overlapping shading (Figs. 2b, 7c, 7d) render on different systems (PC vs. Mac). In the new version of the manuscript, we hopefully resolved this problem.

Line 311: Is the narrower seasonality of SSWs in part by construction? Since they cannot be defined past the end of March in the CP07 definition. For a better comparison, you could consider a similar definition that uses the full calendar year- see Butler and Gerber 2018 (this also has the advantage of removing weak March SSWs; which may influence your results on the surface responses to SSWs being slightly weaker).
We did a quick test of what you suggest (Figure R1), showing (left) frequency and (right) surface response as a function of time of the year (0 = 1/1). The top row uses the original definition (of this paper and CP07), the bottom row is for using the full calendar year. The frequency of SSWs

increases from 58% (CP07) to 62% (full calendar year), and the average surface response decreases from 1.7 hPa to 1.55 hPa. The narrow seasonality of SSWs does not change; the frequency of SSWs simply continues to decrease into April (day 90-120). Also, the surface response of these late SSWs is very weak and even negative. However, we did not implement the modified 30-day rule of Butler and Gerber (2018) here, but there are only very few April-SSWs, too few to plot the surface response during the last third of April. The 30-day rule would have reduced this number even further.

[Figure]

Figure R1: (left) Frequency (%) and (right) surface response (hPa) of (black) SSWs, (red) PWDs and (blue) NWDs for (top) using the original time period of this paper (11/1 - 3/31 for SSWs, 11/1 – 5/31 for WDs) (which is just Figs. 7a and 7c) and (bottom) using the full calendar year (7/1-6/30).

Line 319: here, U- events are likely dynamic final warmings, correct? I would distinguish these events from radiative final warmings (this is done to later, on lines 361-365, but perhaps that discussion should be moved up to here).
Thanks, we like this idea, "dynamical and radiative FWs". We introduced this language into section 3.7 of the manuscript.

Line 320-321: I think here you are referring to the fact that the CP07 definition doesn't permit April SSWs, but perhaps that should be made clearer. I would find it hard to believe that the wind would reverse in April and still show a "complete vortex recovery"- since the climatological mean date for the final warming is ~April 12 and after that the winds are climatologically easterly, so what would a complete vortex recovery look like at that time of year? I would rephrase or remove this sentence.
From Figure R1 one can see that the model produces some CP07 SSWs during April, just not very many. Statistically, there occurs one April-SSW every 25 years (f = 4%). Given the internal weather variability, the vortex recovers once in a while after a reversal even when the

climatological FW date has passed. We tried to rephrase this sentence: "Some of the April U-events are associated with a complete vortex recovery (not shown), but these events are not SSWs since they are not permitted by the CP07 definition."

Line 350: There appears to be no significant Pacific response to SSWs so I would remove that from the sentence.
We removed "and also SSWs to some extent".

Could more be said about the NWD response? Intriguingly, the Pacific sector stands out much more strongly for NWDs (and for PWDs), compared to SSWs.
We now write: "The SLP response to NWDs is roughly inverse to that of PWDs, except that the North Pacific SLP anomalies of NWDs are stronger (i.e., +3 hPa vs. -2 hPa). Perhaps, remote forcing from ENSO plays a more important role for NWDs than for PWDs".

Line 350-354: Following the previous comment, could you show these patterns for ENSO-neutral years only as in Figure 4? It seems like for over >4000 samples as in most of the model composites, and *if* the model does not show a strong preference for SSWs/PWDs/NWDs in only one phase of ENSO, then ENSO should basically average out, meaning that Pacific signals shouldn't really be associated with ENSO as suggested here. Is this the case? (i.e., what is the Nino 3.4 composite value for each of these maps?). Could the prominence of the Pacific signal instead be related to the wave driving itself in some way; a fingerprint of the wave driving response? I realize that the intention to investigate this further is mentioned for another paper, but given the strong differences in Pacific response here, I think it would be at least useful to mention the composite Nino 3.4 value in each case.
We first suspected that the strong North Pacific signals of PWDs and NWDs are perhaps signs of precursors or downward propagation. As you may have suspected, we already made some of the ENSO analysis, but we decided to withhold the results from the current paper because of space issues. But here is some of what we found so far: much of what can be seen over the North Pacific is due to ENSO. As shown in Figure R2, taking neutral ENSO composites of SSWs and PWDs creates a weaker and somewhat different North Pacific signal than using the full composites. Interestingly, now the North Pacific precursors of SSWs and PWDs are similar, forming an East-West oriented dipole.
We follow your advice and now mention the composite Nino 3.4 index of the three event types. The composite index indicates that ENSO strongly influences the chances of wave driving events, but not much on CP07 SSWs. We now write: "Perhaps, remote forcing from ENSO plays a more important role for NWDs than for PWDs. This interpretation is supported by the composite Niño 3.4 index of -0.80 K during NWDs and +0.63 K during PWDs. In contrast, the composite Niño 3.4 index during SSWs is only +0.05 K, indicating that in our model ENSO plays almost no role for SSWs. In a separate upcoming paper, we plan to better understand the role of ENSO in influencing polar vortex events and their surface response."

[Figure]

Figure R2: Composite SLP response (hPa) to (left) SSWs, (middle) PWDs, and (right) NWDs during ENSO neutral years.

Lines 350-353: This statement is true but explains the ENSO related precursors to SSWs, not the response to SSWs, which is typically very North Atlantic-centric. The SSW composite in ERA5 in fact shows the opposite anomaly over the N. Pacific (top left Fig 8); this is likely related to ENSO "noise" affecting the composite in the small reanalysis dataset.
We agree with your interpretation and will discuss this in another upcoming paper.

Lines 368-69: This sentence overstates the conclusions/implications of this study. This sentence gives the impression that it's the upward wave driving itself that is the most important source for stratospheric signals at the surface, but of course the wave driving has to first cause a change in the polar stratospheric circulation for it to have any influence. True, I think this study does show that that change does not have to be a full reversal of the winds as for the "major SSW" definition, but I think this sentence does a disservice to the role of SSWs in general, which has been well established in the literature. (The rest of the paragraph clarifies this point, but I would consider rewriting this sentence to better reflect the results of the study; lines 383-384 I think is a much better starting conclusion sentence for what the study shows).
With this sentence we did not intend to give the impression that the wave driving is the most important source for stratospheric signals. We now write: "The results from this paper challenge the general belief that the reversal of the polar vortex associated with major sudden stratospheric warming events (SSWs) is the key physical element for the creation of stratospheric signals at the surface."

Line 372: PWDs had the same frequency as SSWs but by construction of your chosen threshold, which should be clarified here.
Done.

Line 374-375: "just like SSWs, PWDs were preceded by increased amounts of wave activity flux" – this is also by definition
We removed "preceded by increased amounts of wave activity flux,".

Line 377: Here it should perhaps be clarified "half of all PWDs did not concur with major SSWs", since it's very likely that the other half were minor SSWs or final warmings.

We added "major".

Line 378: the more even distribution at least partially comes from PWDs including final warmings, whereas the SSW definition does not include final warmings by construction. While I do think it's an advantage that the PWD definition can detect dynamic final warmings that likely have similar impacts as SSWs, this distinction should be made clearer, as the SSW definition was purposefully designed to exclude final events, so it can't really be considered as a fault of the definition.
This was not intended to be a value statement, just a fact. Anyhow, we now write more precisely: "For example, since our definition of PWDs also permits dynamical final warmings and SSW-like events in April, PWDs were more evenly distributed over the winter than SSWs."

Line 378-379: I don't think this was shown anywhere. This assumes that the Pacific response in Figure 8 is from ENSO, but I think more will have to be provided (such as the composite value of Nino 3.4 for each plot in Fig 8) to make this statement.
Since we now also include a description of the composite Nino 3.4 index in the results section, we think it is justified to leave this statement in the conclusion. We now write somewhat more carefully: "There was also the indication that PWDs are more sensitive to the influences from ENSO than SSWs".

Line 388: I disagree with this point. If there's any reason to use the common SSW definition, it's for its simplicity, particularly in model data because you just need daily u1060. It's much more intensive to calculate Fz (for which you need gridded daily v and T). I think a stronger stance this study could take is that you've demonstrated that PWDs are valuable measures of stratosphere-troposphere coupling, so there should be more demand for models to output daily v'T' (pre-calculated preferably on model time steps), and to participate in efforts to output these dynamical variables like DynVarMIP (few CMIP6 models did so).
Good point! We eliminated the word "simple" from this sentence, and also provide a discussion of the disadvantages of our definition. We write: "On the other hand, there are also disadvantages to the PWD definition. First, it requires knowledge of EP-fluxes, which are more complicated to calculate than the simple zonal mean zonal wind for SSWs. In addition, EP-fluxes are often unavailable from models, highlighting the need to make these and other dynamical variables publicly available to projects like DynVarMIP (Gerber and Manzini, 2016). Lastly, for certain applications, it may be a disadvantage that PWDs do not distinguish between different event types. However, some of this information can be easily added, as done in this study for U- and U+ events."

Additionally, while PWDs may include all these types of events, they don't distinguish between them, which is not necessarily an advantage. Instead, perhaps this should be stated as, while other definitions focus on separately defining all these variations of polar vortex variability and looking at differences between them, this goal of this definition is to identify events with greater surface impact, no matter the particular timing or evolution of the polar stratospheric circulation.
See our response to your previous comment.

Line 391: I don't think this can be stated so strongly here unless you test this explicitly. I agree that Figure 3b hints some lengthening of wave activity flux prior to the event, but it's not a guarantee that this will translate into better predictability of these events. I would soften this statement, e.g., "potentially lengthen the forecast horizon"…

Yes, agreed, we changed our wording.

**Technical Corrections**

Line 19, line 56, and throughout: capitalize Southern/Northern Hemisphere
Corrected

Line 50: I don't think NAM has yet been defined anywhere
Corrected

Line 55-56: change to "stronger climatological polar vortex"
Corrected

Line 97: change "date" to "dataset"
Corrected

Line 106: change to "The results in Section 3…"
Corrected

Line 145: "past the wave driving" – remove "the"
Corrected

Line 157: "As for SSWs"- suggest instead "Similarly to SSWs"
Corrected

Line 233, line 360: don't need to capitalize "polar cap"
Corrected

**RC2**

**General Comments**

This is an interesting study that aims to introduce another definition of extreme polar vortex events, related to the magnitude of the lower-stratospheric wave flux. The authors compare the tropospheric response to the most-often used wind-reversal criterion, to their own wave driving definition and find that the latter gives an overall stronger surface response. Although the suggested definition does have some advantages as the authors state, I do not think it is as simple to calculate as the wind reversal criterion or indeed other definitions that require a single zonal-mean field. In some places therefore, the language should be toned down so as to not over-sell this new definition.

We tried to change our language throughout the whole manuscript. For example, we now describe in the conclusion section also some of the disadvantages of the wave flux definition.

A bugging concern I have is that the presented diagnostic is not necessarily capturing a wave driving event, rather an increased wave flux in the lower stratosphere. So the nomenclature should be changed from planetary wave driving (PWD) to planetary wave flux (PWF) events. Even though there is a strong EP flux in the lower stratosphere, it may not drive a weaker vortex and the wave activity could just propagate and break farther equatorward. Indeed, it is the derivative of the wave flux that determines how much the mean flow is affected, and so this should be called the wave driving, rather than your definition.

We agree of course. It is the convergence of the flux and not the flux itself what matters. In the literature, however, the term "wave driving" is often used to refer somewhat incorrectly to the wave activity flux. This probably goes back to Newman and Nash (2000) and Newman et al. (2001), where they for example write:

> "The zonal-mean eddy heat flux is directly proportional to the wave activity that propagates from the troposphere into the stratosphere. This quantity is a simple eddy diagnostic which is calculated from conventional meteorological analyses. Because this "wave driving" of the stratosphere has a strong impact on the stratospheric temperature, it is necessary to compare the impact of the flux with respect to stratospheric radiative changes caused by greenhouse gas changes." (Newman and Nash, 2000).

Also, as we show in Figure 6, most of the lower stratospheric wave flux leads to a deceleration of the vortex and is thus absorbed by the vortex, indicating that wave flux and its convergence are very closely related. For these reasons, we prefer to not change the use of "wave driving" in our paper. However, we now point out in a footnote the somewhat incorrect terminology. We write: "The term wave driving is perhaps somewhat misleading because it is the convergence of the wave flux and not the flux itself that drives the polar vortex. However, in the literature, stratospheric wave driving is often used to refer to the flux, and we keep with this tradition".

I would like to see how the presented definition compares to a more dynamical extreme vortex event definition such as the wind tendency definition of Birner and Albers (2017; SOLAS). In my eyes, that definition can more appropriately capture wave driving events, as the wind deceleration is proportional to the wave flux convergence (in the transformed Eulerian mean sense). Nevertheless, the current study is already long enough and self-contained and so this is a suggestion for future work.

You are probably right. The wind tendency is a more direct measure of the changes to the vortex than the wave or heat flux. But using the wind tendency may also be a disadvantage since it requires knowledge about the vortex itself and not about precursor events that force the vortex. In future work we will keep in mind your suggestion and perform a comparison. Perhaps, this may lead to another interesting paper.

The paper is well-written and well thought out and so my comments are generally of a minor nature. Hence, my overall recommendation is of publication subject to minor corrections which I list below.

**Comments**

Lines 44-45; It is relatively well understood now that the wave-mean flow interactions associated with the critical layer mechanism for downward propagation (that originally propsed by Matsuno) only reaches the tropopause. See for instance, Hitchcock and Haynes (2016; GRL). So I would rephrase or remove this sentence.

We think it is important to keep this sentence because one of the key issues addressed by this paper is how relevant the zero-crossing of the vortex is for the surface response. We replaced "downward influence on the surface" with "downward influence on the troposphere".

Lines 51-55: I think the following paper should be cited here: "Defining Sudden Stratospheric Warming in Climate Models: Accounting for Biases in Model Climatologies" by Kim et al. 2017, J. Clim. They make this point about the fixed threshold not being ideal for climate models as there are mean state biases present so that a model with a too strong vortex would likely simulate less SSWs. This is a sort of similar point to that regarding NH vs SH differences.

We now added this sentence. "In addition, the frequency of SSWs simulated by a model is likely to be affected by biases in the strength of the polar vortex if a fixed threshold criterion is used (e.g., Kim et al., 2017)."

Lines 66-67; The heat flux itself is not referred to as the wave driving, as what happens if the wave activity simply propagates upward through a region? Rather, it is the derivative of the heat flux that weakens the polar vortex as this represents the convergence/divergence of said wave activity.

We commented on this point in the General Comments section.

General comment on introduction: It is currently very long and I would shorten it to be more to the point. Also, I think other definitions of extreme vortex events should be mentioned somewhere. There are many but the current intro only focusses on the wind-reversal one. How

does yours fit into the context of others? I would add a paragraph to discuss previous definitions. Tendency based definitions such as that by Birner and Albers (2017; SOLAS) is one example that may be better suited to overcome the issue you mention on defining SSWs in a warming world with underlying polar vortex changes (your lines 58-60). Sentences such as that on lines 62-64 make it seem that your definition is the only study that has attempted to use a more dynamically-based definition.

We removed the sentence on lines 62-42. We also added some more discussion on other SSW definitions: "Different methods have been proposed for the detection of polar vortex events; the papers by Palmeiro et al. (2015) and Butler et al. (2015) give excellent overviews. Most of the methods have in common that some property (e.g., temperature, zonal wind, or geopotential height) of the polar vortex is used, either in terms of an absolute threshold, a pattern, a gradient, or a tendency. Birner and Albers (2017), for example, use the tendency of the zonal mean flow to better capture "the explosive dynamics of these events"."
We also tried to shorten some of the introduction.

Lins 125-126: I think the WMO criterion also involves a reversal of the temperature gradient. The CP07 definition is a simplification of that.
We now no longer use the expression "WMO-definition".

Lines 139-144: Is there a reason an e-folding timescale of 50 days is chosen? Is this something to do with radiative timescales in the lower stratosphere? Or perhaps related to the 40-day vertically integrated wave flux in Polvani and Waugh (2004)? It would be good to know if results are sensitive to varying this parameter to shorter timescales (which would be a more conservative criterion that reduces the accumulated wave flux). Intuitively 50 days sounds quite long, so would be good to have justification here.
Choosing a long enough time scale is crucial. We experimented with a range (ca. 10-60 days) of periods and found that using shorter periods results in sharper events. But these sharper events then tend to be preceded (~ at day -30) by stronger positive stratospheric NAM anomalies (shown in blue in Figure R3) and are followed by a weaker surface impact after onset. To illustrate this, we show in the following figure the composite NAM anomalies of PWDs from using a 30-day-long tau.

[Figure]

Figure R3: As Figure 3j of the original manuscript but using 30 instead of 50 days to average the upward wave activity flux at 100 hPa.

Also, Newman et al. (2001) and Polvani and Waugh (2004) used similarly long 40-45 day time scales. A related question is what sets the ~30 day long time between the preceding positive

and following negative NAM anomalies, which can be seen in both SSWs and PWDs. The NH annular mode decorrelation time scale (Gerber 2008) is also on the order of ~30 days, and also the time it takes for the NAM anomalies after events to die out. We suspect that the 30 day period is the characteristic time scale of stratospheric vacillations, and as you say, this may be in part influenced by the radiative timescale in the (lower) stratosphere. Other effects like wave propagation may also play a role. In summary, it seems tau should be long enough (> 30 days) as to suppress the selection of natural vacillations that are preceded by negative upward directed wave activity fluxes.

We now added the following explanation: "By experimentation we found that it is important to use a long enough e-folding time $\tau$: a shorter $\tau$ selects events that tend to be preceded by stronger negative stratospheric NAM anomalies and followed by weaker surface responses. The $\tau$ = 50 day of our study is also similar to previous studies (Polvani and Waugh, 2004; Newman et al., 2001)."

Lines 148-150: 1) What does the timescale physically mean? The time taken for an average accumulated wave packet to die out?
Please see the answers to your previous question.

2) Why is a negative Fz set as the end of the wave driving event? I would think that if the wave flux anomaly was close to zero but still positive then this is pretty much the end of the event anyway. For instance, imagine a situation where the wave flux anomaly remained positive but close to zero for an extended period; this would be erroneously counted as an extended event and contribute to the summation. Would a more plausible end of event be related to a criterion on the standard deviations?
Yes, such a situation may occur, but in practice it is not very likely. Scaling by the standard deviation is a good idea, but then one still faces the problem to define what fraction of standard deviation is "close to zero". We essentially chose a threshold of zero, and as long as the wave flux is positive it contributes to the summation. As our papers demonstrates, this seems to be working quite well. Also, maybe there is some misunderstanding: the criterion only waits until the daily Fz anomaly turns negative, and not the accumulated Fz.

Lines 162-164: Physically a positive PWD in the 20 days after an SSW is surely not related to the driving of the SSW? Indeed, Fz remains positive for 2-3 days after an SSW event (as your figure 3 shows), but for 1 week plus, I highly doubt it. How many of your common events fall into this category of large Fz in the 20 days after an identified SSW?
We wanted to keep things simple, and the maximum 20 day separation period seemed reasonable given that a similar time is used as criterion for multiple SSWs. Do you think that the 20 day window is too long, and that some of the PWDs that follow SSWs are erroneously counted as identical? But any (perhaps asymmetrical) threshold will always be somewhat arbitrary. Also, the 2-3 days you mention above is just an average over many events, and it is easy to imagine more complicated situations. We are also talking about a small number of events, as the histogram in Figure R4 shows. Given Figure R4, a separation distance of 20 days seems reasonable, but 10 days would have been another good choice.

[Figure]

Figure R4: Histogram of the separation distance d (days) of common events (date$_{SSW}$ - date$_{PWD}$) in the model.

Line 189: What is this 58% and 62% SSW frequency? I presume you are referring to the number per decade which in ERA-5 would be 6.2 per decade and so translates to 62% of years having an SSW. Is that right? I am not saying the statistics are wrong, rather the way they are presented is non-standard.
To clarify this, we now added: "Here and in the rest of this paper, the event frequency is given in %, i.e., events per year multiplied by 100." Also, we think it is quite common to give the frequency as events per year, e.g., SPARC-CCMVal (2010).

Figure 2: I am not sure if it is to do with the way it is rendered on my screen, but the shading in this figure completely masks much of the figure. It not only masks the lines, but the writing next to the lines. Please fix and make the confidence interval more transparent. Because of its current rendering, lines such as 201-202 are impossible to make out.
There is a technical problem in how the figures with overlapping shading (Figs. 2b, 7c, 7d) render on different systems (PC vs. Mac). In the new version of the manuscript, we hopefully resolved this problem.

Figure 3: I cannot distinguish the bold from the non-bold lines in panels a-f. You state that bolded lines represent those differences that are statistically significant.
Maybe there is another technical issue with how the figures upload to the journal and then download and display on your system. We can clearly see the differences between thick and thin lines:

[Figure]

Figure R5

We could draw the thick lines even thicker, but this would have negative consequences for how much detail can be seen from the curves.

Also, the panels b,e are cut off and do not show some of the lines around the onset date. Especially in panel e, the sharp increase in SLP just after day zero is interesting.
We expanded the y-axis plot range for the panels in Fig. 3.

Finally, what is NNR in the caption?
Good catch: NNR (= NCEP/NCAR reanalysis) should be ERA5 and has been corrected.

Lines 216-217: Are you reading off the anomalies in c-d by comparing the solid lines with the dashed lines and seeing which lies lower?
Yes, this is what we do. To make this clearer, we now added: "From the differences between the continuous lines and the dashed lines for the climatology in Figs. 3c-d, one can see that …".

Lines 240-241: From the left column of figure 3, I would not expect to have such comparable SLP anomalies to those in the model. In fact PWDs in ERA5 appear to have positive SLP anomalies, for up to 20-30 days before the onset date, presumably associated with a strong Siberian High.
Figure R6 below shows the results for ERA5. You were right, PWDs and also SSWs to some extent are preceded by strong positive SLP anomalies over Siberia, but somewhat similar features can also be seen in the model ensemble (Figure 4 of the manuscript). In the manuscript, we write "somewhat similar but much noisier patterns", and we hope that this describes the (di-) similarities between model and reanalysis well.

[Figure]

Figure R6: As Figure 4 of the original manuscript, but for the ERA5. Shown are the composite SLP anomalies before event onset of 12 PWDs and 12 SSWs during neutral ENSO years.

Lines 267-270: 100hPa where you identify the PWDs is already well within the vortex. Hence, the PWDs events you capture may also be due to 'stratospheric internal dynamics'. de la Camara et al. (2017) suggested that 300hPa was a better diagnostic level to say that there is

cross-tropopause wave propagation. Nevertheless, this brings up another point: how sensitive are your PWDs to choice of vertical level? It would be good to raise or lower the level and recalculate the numbers to check that 100hPa is representative of the lower stratosphere.

Yes, agreed, PWDs may also originate from internal stratospheric dynamics because the 100 hPa level is already well above the tropopause. We have chosen 100 hPa as the lower stratosphere simply to be consistent with many previous studies. Testing the sensitivity of PWDs to the level of the wave activity flux is a good idea but would be the subject of another study. Also, changing the level would also mean that the critical threshold for Fz needs to be adjusted, so there is only limited insight from doing so. For the present study, the exact origin of the flux at 100 hPa is not important. In fact, De la Camera et al. (2017) showed that only 1/3 of the wave flux variance at 100 hPa can be explained from the flux at 300 hPa. We added more discussion on this issue in the introduction, where we now write: "We note that the flux at 100 hPa should not be simply interpreted as a wave propagation from the troposphere into the stratosphere. Cámara et al. (2017) showed that only 1/3 of the wave flux variance at 100 hPa can be explained from the flux in the upper troposphere (300 hPa). They argued that the 100 hPa level is well above the extratropical tropopause and is thus already under the considerable influence of stratospheric processes. In the context of our study, however, the exact source for the wave activity flux is less important." We also somewhat adjusted in this section our discussion of what drives SSWs and PWDs (internal dynamics or wave fluxes).

Line 275: The word 'tend' here suggests that the majority of NWDs occur close to the onset date of the SSW (say within +-30 days). I do not see that in figure 5. Rather, there are around as many NWDs that are not close to an onset date as there are close to an onset date. Can you clarify what you mean here, perhaps quantitatively. Otherwise I would just consider removing the sentence.

We think it is quite interesting that many NWDs occur in proximity to weak vortex events. We improved our wording and now write: "Fig. 5 shows that many NWDs occur in close proximity to warm vortex events (e.g., 1981, 1982, 1988, 1995, …), …". We also mention that there are many isolated NWDs.

Lines 331-333: This is interesting but unsatisfyingly not further addressed! Do you have any idea as to why this is? By March-April the vortex is already starting to break down and wave activity to wane (figure 1a,b) and so is the weaker day 0-59 SLP response simply reflecting the seasonal cycle? The vortex recovery is too weak by this point as it is the transition time to easterlies and so the radiative recovery is cut off by the seasonal cycle.

We think you provided the answer to your question. In the paragraph just before this section and at various other places of our manuscript (introduction, section 3.4, section 3.6, conclusion) we mention that the CP07 definition includes late and dynamically not very active SSWs, with a climatological vortex that is weak, and breaking it down does not create a strong surface response. To make this clearer, we now write here: "The likely reason is that dynamically these events are not very active; the climatological vortex is weak during this time of the year and small amounts of wave activity suffice to trigger the SSW criterion.".

Many studies have shown that the persistent lower-stratospheric anomalies are important for a continued tropospheric response (Hitchcock and Simpson 2014, JAS; Maycock and Hitchcock 2015, GRL; White et al. 2020, JClim) with it being mechanistically attributed to the induced meridional circulation by the lower stratospheric radiative recovery (Thompson et al. 2006, JAS; White et al. 2022, JAS), and this may provide further evidence for that. A NAM index plot for SSWs occurring only in March-April would help to see if the extended recovery in the lower stratosphere is indeed cut off by the seasonal cycle with a shorter NAM timescale evidence for that.

Thank you for your interpretation. Figure R7 below shows the requested plot. The u1060 plots show that late SSWs quickly transition into climatology without an extended recover period, whereas PWDs show a much longer recovery period. It can also be seen that late PWDs are essentially FWs, that late SSWs are associated with little Fz, and that the NAM perturbations after late SSWs are much weaker than that of late PWDs. We still believe that a strong preceding wave activity flux and strong vortex perturbation is the simplest explanation for creating active events with a strong surface response.

[Figure]

Figure R7: As Figure 3 of the original manuscript, but for late (March or later) events only. The composites contain (left) 1544 SSWs and (right) 1025 PWDs.

Figure 7: Same problem as figure 2 with the shading.
Yes, and we tried to resolve this in the revised version of the manuscript.

Lines 350-354: It does not look like the SSWs are associated with a SLP<0 response over the North Pacific compared to the PWDs (comparing panels in column 2) although you state this to be the case.

The negative North Pacific SLP anomalies for SSWs is weak, as shown by the extra dotted contour line at -0.25 hPa. Since the anomalies are statistically not significant, we now removed the words "and also SSWs to some extent".

Isn't the strong SLP<0 anomaly over the North Pacific in the PWDs compared to the SSWs just related to the fact that the PWDs are wave events themselves? As you say, it represents a deeper Aleutian Low but the difference between the PWDs and the SSWs is that the planetary wave driving is shut off in the SSWs whereas it continues in the PWDs until the Fz anomaly goes negative (which could take a while depending on your specified e-folding timescale). Hence, in the PWDs, I would likely expect a more negative Aleutian Low to persist well after the onset date.

Figure 3b demonstrates that SSWs and PWDs have very similar temporal evolutions of Fz and very similar onset dates. Since you mention "until the Fz anomaly goes negative …", we realize that there is maybe some misunderstanding of how we define PWDs. The Fz becoming negative criterion to determine the onset date of a PWD is related to the daily Fz, and not to the accumulated Fz.

To clarify, my concern is that the presented SLP patterns for the PWDs are simply aliasing the planetary wave patterns that drove the weaker vortex in the first place and therefore not part of some downward response.

Aliasing of the wave activity that drove the PWD is unlikely since the onset date of PWDs is defined when the daily Fz turns negative. The negative North Pacific SLP anomalies are also not related to the downward response. Instead, we believe that they are part of El Nino and its associated wave #1 forcing over the North Pacific. Maybe we did not explain this clearly enough, and we therefore write now: "The deepening of the low intensifies the planetary wave #1 activity, provides some of the wave forcing needed for PWDs, and increases the likelihood for polar vortex events  (Garfinkel and Hartmann, 2008)."

Perhaps a simple way around this is not to use such a broad time-average window that goes all the way to the onset date (i.e., not use lags 0-59). Or, base the averaging window on the date the minimum stratospheric winds were found following the maximum Fz.

From analysis which we do not show in this paper, we find that ENSO largely influences the occurrence of PWDs and NWDs. This provides the best explanation for the negative North Pacific SLP anomalies after PWDs. For example, the ENSO influence is demonstrated from the composite ENSO index during the different event types. We now mention this in this section by writing: "… the composite Niño 3.4 index of -0.80 K during NWDs and +0.63 K during PWDs. In contrast, the composite Niño 3.4 index during SSWs is only +0.05 K, indicating that in our model ENSO plays almost not role for SSWs." Also, Figure 4 of the original paper shows SSW and PWD composites for neutral years. Looking at these composites for times after onset (not shown in Figure 4) one finds that the negative North Pacific SLPs after PWD onset becomes weaker (ca. 1 hPa for neutral ENSO vs. 2 hPa for all ENSO). See Figure R8 below.

[Figure]

Figure R8: The composite SLP response after (day 0-59) the onset of (left) SSWs, (middle) PWDs and (right) NWDs during neutral ENSO years.

Further, I thought you had removed the ENSO effects from the timeseries (lines 174-179). Please clarify as this affects the discussion here. It also affects for instance, line 359.
Lines 174-179 only explain how we calculate ENSO. Most of our figures include ENSO-effects and are for all years. Only for Figure 4 we removed ENSO by focusing on events during neutral ENSO years. We now emphasize this point at the beginning of section 3.7.

Line 388: I don't think this is such a simple metric to calculate, particularly compared to the wind reversal one. The traditional definition can capture most of these events. I would not state that your definition trumps it so flippantly.
We agree and remove the word "simple". We also add to the conclusion some discussion of some of the disadvantages of the PWD definition.

Line 391: Compared to the traditional measure, it appears that the PWD events hint at around an extra week of extreme Fz (figure 3b) but I would hesitate to state that this so definitely.
We agree, it remains to be seen whether and by how much the PWD definition lengthens the forecast horizon. But there is the potential to do so. We remove the words "by several weeks" and add "may" at the beginning.

**Technical Comments**

Title: You use 'sudden' twice!
Corrected

Line 18: 'criterium' --> 'criterion'
Corrected

Line 19: Duplicated 'that'
Corrected

Line 97: 'data' --> 'data'
Corrected

Line 106: Change the short sentence to 'The results in Section 3...'
Corrected

Line 134: To clarify, did you use area weighting to average the EP flux?
Yes, and we now clarify this in the text

Figure 1: I think I get it, but can you clarify what the lines mean? I presume thin lines represent the +-SDs and the thick represents the full value, but it would be nice to not have to work it out...
We now make this clearer in the figure caption

Line 146: remove the first 'the' on this line.
Corrected

**References**

Birner, T. and Albers, J. R.: Sudden stratospheric warmings and anomalous upward wave activity flux, SOLA, 13A, , 8–12, doi:10.2151/sola.13A-002, 2017.

Black, R. X. and McDaniel, B. A.: The Dynamics of Northern Hemisphere Stratospheric Final Warming Events, Journal of the Atmospheric Sciences, 64, 2932-2946, 10.1175/jas3981.1, 2007.

Butler, A. H., Seidel, D. J., Hardiman, S. C., Butchart, N., Birner, T., and Match, A.: Defining Sudden Stratospheric Warmings, Bulletin of the American Meteorological Society, 96, 1913-1928, 10.1175/bams-d-13-00173.1, 2015.

Cámara, A. d. l., Albers, J. R., Birner, T., Garcia, R. R., Hitchcock, P., Kinnison, D. E., and Smith, A. K.: Sensitivity of Sudden Stratospheric Warmings to Previous Stratospheric Conditions, Journal of the Atmospheric Sciences, 74, 2857-2877, 10.1175/jas-d-17-0136.1, 2017.

Garfinkel, C. I. and Hartmann, D. L.: Different ENSO teleconnections and their effects on the stratospheric polar vortex, Journal of Geophysical Research: Atmospheres, 113, D18114, 10.1029/2008JD009920, 2008.

Gerber, E. P. and Manzini, E.: The Dynamics and Variability Model Intercomparison Project (DynVarMIP) for CMIP6: assessing the stratosphere–troposphere system, Geosci. Model Dev., 9, 3413-3425, 10.5194/gmd-9-3413-2016, 2016.

Newman, P. A. and Nash, E. R.: Quantifying the wave driving of the stratosphere, Journal of Geophysical Research: Atmospheres, 105, 12485-12497, https://doi.org/10.1029/1999JD901191, 2000.

Newman, P. A., Nash, E. R., and Rosenfield, J. E.: What controls the temperature of the Arctic stratosphere during the spring?, J. Geophys. Res., 106, 19999-20010, 19910.11029/12000JD000061, 2001.

Palmeiro, F. M., Barriopedro, D., García-Herrera, R., and Calvo, N.: Comparing Sudden Stratospheric Warming Definitions in Reanalysis Data, Journal of Climate, 28, 6823-6840, 10.1175/jcli-d-15-0004.1, 2015.

Polvani, L. M. and Waugh, D. W.: Upward wave activity flux as precursor to extreme stratospheric events and subsequent anomalous surface weather regimes, Journal of Climate, 17, 3548-3554, 2004.

SPARC-CCMVal: SPARC Report on the Evaluation of Chemistry-Climate Models, WCRP-132, WMO/TD-No. 1526, 109-148, 2010.

---

## Author Response (AR2)

**RC1**

I thank the authors for addressing the majority of my comments. There is now a better inclusion and discussion of NWD events in the study. The tone regarding the SSW definition is also improved.

I only have a few remaining minor/technical comments. Line numbers here are based on the "tracked changes" version of the manuscript.

Line 20: perhaps "lengthens the potential prediction horizon of the surface response" since changes in the predictive skill have not actually been evaluated here

See response to next comment.

Line 21: This last point regarding the SH is an interesting one, but it's not the focus of the study and is now only briefly brought up at the very end of the paper, so it seems out of place to be brought up as a main point in the abstract. I wonder if this list of PWD advantages in the abstract should more closely echo the bullet points in the Conclusions.

We now write: "… the wave activity flux definition captures with one criterion a variety of different event types, detects strong SSWs and strong final warming events, avoids weak SSWs that have little surface impact, and potentially lengthens the prediction horizon of the surface response."

Line 231: change to "ERA5 leads"

Done.

Line 253: change to "zero crossing to 80-90 days after the onset"

We now write: "As can be seen from the time of the $u_{1060}$ zero crossing at 80-90 days after the onset of SSWs and PWDs, …"

Line 258: I don't see an "under-recovery" following NWDs, the line looks almost overlying the climatological line.

Looking closely one can see that the blue line is somewhat below the blue dashed line. We therefore write: "Afterwards, there is a slight "under-recovery" of the vortex and the FW date of NWDs is about normal."

Line 283: change "negative one" to "negative anomaly"

Done.

Line 284-287: I would delete this here since you talk about the ENSO relationship later on. It's somewhat counterintuitive to talk about the composite ENSO value in plots that are

supposed to be ENSO-neutral by construction. Plus, I'm not sure it's needed here (or the additional text about composite Niño 3.4 index values in the caption of Figure 4). If you do keep it, change "explaining" in line 287 to "explain".

It seems we did not explain this very well. The composites are for ENSO-neutral years only, but still the composite ENSO index is somewhat negative. We think it is important to note this here, but now write hopefully somewhat clearer: "Even though only events from neutral ENSO years are included in these composites, the composite Nino 3.4 index for NWDs during these neutral ENSO years is -0.24 K. This suggests that NWDs are somewhat favored by La Nina-like conditions, which explains the persistent positive SLP anomalies over the North Pacific that start long before onset."

Line 326: "also" is redundant to "In addition" here

We now removed "also".

Line 335: change to "March 2016". You should also mention that this was a final warming (the wind did reverse, but then stayed easterly so that's why it doesn't show up as a SSW).

Done.

Line 341: "warm vortex events" – here it's not clear what is meant by this new terminology. Are you referring to PWDs? SSWs? Both? From the dates provided it seems likely you mean PWDs.

We changed "warm" to "weak".

Figure 6 and discussion on lines 359-379: I realize the many samples here imply the small mean shifts in these distributions are still significant. However, I wonder if you could apply a statistical test to see whether these distributions are actually significantly different. For example a Kolmogorov-Smirnov distribution test. If anything to me Figure 6 actually suggests little difference in the distribution of 0-59 day SLP response after SSWs vs PWDs, even if the mean is slightly shifted. Without some statistical assessment I'm not sure this figure supports your argument of a more robust response for PWDs.

Figure 6 is not meant to support our argument of a more robust for PWDs. In discussing this Figure, we simply say: "Compared to SSWs, PWDs create on average a somewhat stronger mean response (2.0 hPa vs. 1.7 hPa), reduced response spread (3.5 vs. 3.6 hPa), and reduced chance of a negative $\overline{slp}^{PC,0-59}$ (29% vs. 32%)". Thus, we do not claim that the difference between the two distributions is statistically significant. However, from Figure 2b (top), and as also mentioned in the discussion of Figure 2b (section 3.2), one can see that for FZ=12.9 days the 95% confidence intervals of the surface response for SSWs and PWDs are well separated from each other. This indicates that PWDs lead on average to a significantly stronger surface response than SSWs.

Line 432: suggest changing "polar vortex events" to "PWDs", since there appears no deepening of the Aleutian low in the SSW panel (and SSWs are also polar vortex events)

Done.

Line 433: change to "except that the magnitude of the North Pacific SLP anomalies"
Done.

Figure 9 does not seem very necessary, given that you could convey the point of the figure in words, and Figure 5 shows something similar (but is more useful since it compares to SSWs as well).

We agree and remove Figure 9 and its discussion from the new manuscript.

---

## Author Response (AR3)

**Co-Editor Comments**

We made the three changes suggested by the co-editor, but I don't understand the comments regarding the corresponding author(s). As indicated in the manuscript, the paper has one corresponding author (Thomas Reichler).